# Align Your Trajectory Tangent: Training Better Consistency Models via Manifold-Aligned Trajectory Tangents

Beomsu Kim [* 1]  Byunghee Cha [* 1]  Jong Chul Ye [1]

## Abstract

With diffusion and flow matching models achieving state-of-the-art generating performance, the interest of the community now turned to reducing the inference time without sacrificing sample quality. Consistency Models (CMs), which are trained to be consistent on diffusion or probability flow ordinary differential equation (PF-ODE) trajectories, enable one or two-step flow or diffusion sampling. However, CMs typically require prolonged training with large batch sizes to obtain competitive sample quality. In this paper, we examine the training dynamics of CMs near convergence and discover that CM trajectory tangents – CM output update directions – are quite oscillatory, in the sense that they move parallel to the data manifold, not towards the manifold. To mitigate oscillatory trajectory tangents, we propose a new loss function, called the *manifold feature distance (MFD)*, which provides manifold-aligned trajectory tangents that point toward the data manifold. Consequently, our method – dubbed *Align Your Trajectory Tangent (AYT)* – can accelerate CM training by orders of magnitude and even out-perform the learned perceptual image patch similarity metric (LPIPS). Furthermore, we find that our loss enables training with extremely small batch sizes without compromising sample quality. Code is available at https://github.com/1202kbs/AYT

## 1. Introduction

Diffusion models (DM) (Sohl-Dickstein et al., 2015; Ho et al., 2020; Song et al., 2021a) and flow models (FM) (Liu et al., 2022; Liu, 2022; Lipman et al., 2023) have achieved remarkable progress in generative modeling over the past few years. Their strength lies in their ability to trade-off sample quality with sampling cost. Concretely, by increasing the number of score model or velocity evaluations during sample synthesis, one can reduce error in solving diffusion SDEs or flow ODEs, and thus enhance the quality of synthesized samples (Song et al., 2021b; Lipman et al., 2023). With DMs and FMs achieving state-of-the-art generative performance (Dhariwal & Nichol, 2021; Karras et al., 2022; 2023; Esser et al., 2023), the interest of the community turned to reducing the inference cost without compromising sample quality (Lu et al., 2022; Dockhorn et al., 2022; Salimans & Ho, 2022; Zhang & Chen, 2023; Kim & Ye, 2023).

One promising learning-based approach to accelerating DMs and FMs is consistency models (CM) (Song et al., 2023), which are trained to transport noise to data along PF-ODE trajectories with only a minimal number of, *e.g.*, one or two, neural net evaluations. However, CM learning is often unstable, and is prone to divergence during training (Song et al., 2023). Subsequent works have found that better hyper-parameter choices (Song & Dhariwal, 2024; Lu & Song, 2025), techniques such as truncation (Lee et al., 2025), or joint learning of diffusion score or flow velocity (Kim et al., 2024a; Boffi et al., 2025; Geng et al., 2025a; Sabour et al., 2025) can accelerate and stabilize training.

In this paper, we take an orthogonal, training dynamics-based approach to improving CMs by examining and enhancing CM loss functions. While there are works which propose better loss functions for DM or FM training (Hoogeboom et al., 2023; Kim et al., 2025a; Lin & Yang, 2024; Berrada et al., 2025), losses for CMs have been left relatively under-explored after the pseudo-Huber loss gained popularity due to its ability to reduce variance during training (Song & Dhariwal, 2024). The learned perceptual image patch similarity (LPIPS) metric (Zhang et al., 2018) has shown to be a powerful loss for training CMs (Song et al., 2023; Kim et al., 2024a), but construction of LPIPS involves extensive engineering such as supervised pre-training on ImageNet (Deng et al., 2009) and fine-tuning on a human-curated dataset of patch similarity.

Given such situation, in our work, we propose a loss function which is as powerful as LPIPS for training CMs, but

---

[1]Graduate School of AI, KAIST. *Equal Contribution. Correspondence to: Jong Chul Ye <jong.ye@kaist.ac.kr>.

*Proceedings of the $43^{rd}$ International Conference on Machine Learning*, Seoul, South Korea. PMLR 306, 2026. Copyright 2026 by the author(s).

is simpler to construct and does not require human supervision. Unlike pseudo-Huber or LPIPS, our loss is grounded on a rigorous analysis of CM training dynamics, such that it consists of interpretable design choices. We test our loss function on a number of variety of image generation tasks, and show that it accelerates CM training by orders of magnitude when compared to training with the pseudo-Huber loss. Furthermore, our loss simultaneously improves generative performance, and even outperforms LPIPS.

Concretely, our contributions can be summarized as:

- **We perform a rigorous analysis of slow convergence in CM training (Section 4).** We examine the training dynamics of CMs near convergence and identify that trajectory tangents[1], *i.e.*, update directions for CM outputs, contain non-trivial amount of components which oscillate parallel to the data manifold. We hypothesize that such oscillatory components can hinder the convergence of CMs, and that one must amplify manifold-orthogonal components, *i.e.*, components which point towards the data manifold, to enhance performance.

- **We propose the manifold feature distance to accelerate convergence (Section 5).** We discover that when CM loss is computed in a feature space, trajectory tangents are linear combinations of the rows of the feature map Jacobian. Inspired by this, we design manifold feature maps whose Jacobian consists of directions that point toward the data manifold. Consequently, computing consistency losses with our manifold feature distance provides manifold-aligned trajectory tangents with minimal oscillatory components.

- **We verify our method on a number of benchmark tasks (Section 6).** We train CMs on standard benchmark datasets CIFAR10 and ImageNet $64 \times 64$ with our manifold feature distance. We observe that our loss accelerates convergence by orders of magnitude compared to training with the pseudo-huber loss, and beats LPIPS. Furthermore, training with manifold feature distance is robust to batch size, yielding competitive FID scores with batch size as small as 16. Experiments corroborate our hypothesis that oscillatory components in trajectory tangents hinder CM convergence.

## 2. Related Works

**Regularization for DMs and CMs.** Similar to GANs, regularization for efficient training has also been actively explored in diffusion and consistency models. In particular, early stopping has been introduced to mitigate overfitting,

---

[1]We explicitly use the term "trajectory tangent" to distinguish it from the manifold tangent, which refers to the tangent plane of the manifold.

which often occurs at small timesteps (Nichol & Dhariwal, 2021; Lee et al., 2025). Although simple, this approach is effective in preventing the model from over-adapting to the data distribution and has therefore received significant attention. Subsequently, various data augmentation strategies have been proposed. For example, non-leaking augmentation (Karras et al., 2022) and noise perturbation techniques (Daras et al., 2025; Ning et al., 2023) have been employed to improve generalization and enhance robustness under diverse conditions. More recently, research has shifted toward incorporating more sophisticated auxiliary learning signals. For instance, contrastive learning objectives have been adopted to encourage the model to acquire more discriminative image samples (Stoica et al., 2025), while the outputs of pre-trained representation models have been aligned with intermediate diffusion features to accelerate training and improve convergence stability (Yu et al., 2025; Jeong et al., 2025; Chefer et al., 2025).

**Perceptual objectives.** Various studies have explored the use of perceptual metrics to facilitate the training of diffusion and consistency models. However, due to the nature of score matching, directly minimizing perceptual metrics can adversely affect the training of diffusion models. To address this, some methods use perceptual losses only after the diffusion model has been pretrained (Lin & Yang, 2024), while others incorporate them as auxiliary losses during training (Berrada et al., 2025). In the case of consistency models, perceptual metrics such as LPIPS can be directly employed as consistency losses without compromising theoretical guarantees (Song et al., 2023).

**Fast sampling of diffusion and flow models.** While diffusion models have demonstrated remarkable performance in image and video generation, their sampling process often requires hundreds to thousands of steps, resulting in significant computational cost. To address this limitation, a wide range of approaches have been proposed to enable fast sampling, where high-quality samples can be generated with only a few steps. Early studies (Lu et al., 2022; Zhang & Chen, 2023; Dockhorn et al., 2022; Zhou et al., 2024b) primarily focused on improving ODE solvers. By mitigating error accumulation across timesteps, these methods reduced the required number of sampling steps to about 10. Beyond solver improvements, several approaches have aimed to directly train models capable of efficient sampling. A representative example is Rectified Flow (Liu et al., 2022; Liu, 2022; Zhu et al., 2024; Liu et al., 2024; Lee et al., 2024; Kim et al., 2025a), which straightens the ODE trajectory from noise to image, thereby minimizing error accumulation under a small number of steps. Another line of research is diffusion model distillation (Salimans & Ho, 2022; Meng et al., 2023; Kim et al., 2024b). In this paradigm, a pretrained diffusion model is distilled into a new single-step generative model by leveraging objectives such as diffusion

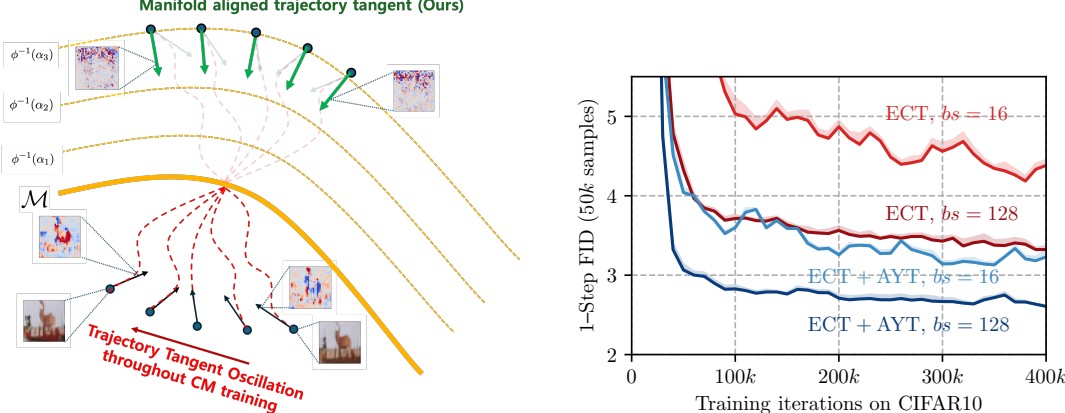

Figure 1. **Left:** CM trajectory tangents, *i.e.*, CM output update directions, exhibit large oscillations throughout training. To mitigate this, we learn feature maps $\phi$ whose level sets $\phi^{-1}(\alpha)$ model increasingly perturbed data manifolds, so feature map gradients point towards the manifold. CM trajectory tangents in the feature space are expressed as linear combinations of feature map gradients, so we obtain manifold-aligned trajectory tangents. **Right:** *Manifold-aligned trajectory tangents (AYT) enable up to $\times 10$ faster convergence and competitive FIDs with $\times 1/8$ batch size (bs).* We use Easy Consistency Training (ECT) (Geng et al., 2025b). Shaded regions indicate min/max FIDs over three sample generation trials.

losses. More recently, flow map–based approaches (Song et al., 2023; Kim et al., 2024a; 2025b; Sabour et al., 2025; Geng et al., 2025a) have been introduced, which learn the trajectory of an ODE directly, predicting the destination at a target timestep from an input at a source timestep. These methods are particularly notable as they can be applied both in the context of distillation and from-scratch training. Consistency models can be interpreted as a special case of this family, where the target timestep is set to zero, allowing the model to predict the final image directly from noise.

## 3. Background

Our goal is to learn a generative model of a data distribution $p(\boldsymbol{x})$ supported on $\mathbb{R}^d$. Given a forward or corruption process from data $\boldsymbol{x} \sim p(\boldsymbol{x})$ to noise $\boldsymbol{\epsilon} \sim \mathcal{N}(\mathbf{0}, \boldsymbol{I})$, define

$$\boldsymbol{x}_t = \alpha_t \boldsymbol{x} + \sigma_t \boldsymbol{\epsilon} \quad (1)$$

for $t \in [0, \infty)$ with boundary conditions $\boldsymbol{x}_0 = \boldsymbol{x}$ and $\lim_{t \to \infty} \alpha_t / \sigma_t = 0$. Additionally, the distribution of $\boldsymbol{x}_t$ at time $t$ is defined as $p_t$. The corresponding probability flow ordinary differential equation (PF-ODE), *i.e.*, an ODE whose marginal equals $p_t$ for all time $t$ is then given by

$$d\boldsymbol{x}_t = \mathbb{E}_{\boldsymbol{x}, \boldsymbol{\epsilon} | \boldsymbol{x}_t}[\dot{\boldsymbol{x}}_t] \, dt = \mathbb{E}_{\boldsymbol{x}, \boldsymbol{\epsilon} | \boldsymbol{x}_t}[\dot{\alpha}_t \boldsymbol{x} + \dot{\sigma}_t \boldsymbol{\epsilon}] \, dt \quad (2)$$

and velocity can be learned by solving flow matching (Lipman et al., 2023; Albergo et al., 2023)

$$\min_{\boldsymbol{v}} \mathbb{E}_{\boldsymbol{x}, \boldsymbol{\epsilon}, t}[\|(\dot{\alpha}_t \boldsymbol{x} + \dot{\sigma}_t \boldsymbol{\epsilon}) - \boldsymbol{v}(\boldsymbol{x}_t, t)\|_2^2]. \quad (3)$$

Let $p_0 = p$ and $p_T \approx \mathcal{N}(\mathbf{0}, \sigma_T^2 \boldsymbol{I})$ for a sufficiently large time $T$, so we can sample from $p$ by sampling $\boldsymbol{\epsilon} \sim \mathcal{N}(\mathbf{0}, \boldsymbol{I})$ and solving the PF-ODE down from time $t = T$ to $0$ with

terminal condition $\boldsymbol{x}_T = \sigma_T \boldsymbol{\epsilon}$. However, numerical integration of the PF-ODE involves multiple evaluations of the velocity, so the generation process is often slow and costly.

A consistency model (CM) $\boldsymbol{f}_{\boldsymbol{\theta}} : \mathbb{R}^d \times [0, \infty) \to \mathbb{R}^d$ with boundary condition $\boldsymbol{f}_{\boldsymbol{\theta}}(\cdot, 0) = \mathrm{id}_{\mathbb{R}^d}$ is trained to be consistent, *i.e.*, to have identical outputs, on PF-ODE trajectories (Song et al., 2023; Song & Dhariwal, 2024; Lu & Song, 2025). Hence, an optimal CM $\boldsymbol{f}_{\boldsymbol{\theta}^*}$ will map all points on the PF-ODE trajectory back to its initial point at $t = 0$ with a single function evaluation. In particular, the output $\boldsymbol{f}_{\boldsymbol{\theta}^*}(\sigma_T \boldsymbol{\epsilon}, T)$ for $\boldsymbol{\epsilon} \sim \mathcal{N}(\mathbf{0}, \boldsymbol{I})$ will be distributed according to $p$, so a CM can bypass the computational burden.

The discrete CM objective (Song et al., 2023) forces $\boldsymbol{f}_{\boldsymbol{\theta}}$ to be consistent on consecutive timesteps[2]:

$$\min_{\boldsymbol{\theta}} \mathbb{E}_{\boldsymbol{x}, \boldsymbol{\epsilon}, t, \Delta t}[(\Delta t)^{-1} d(\boldsymbol{f}_{\boldsymbol{\theta}}(\boldsymbol{x}_t, t), \boldsymbol{f}_{\mathrm{sg}[\boldsymbol{\theta}]}(\boldsymbol{x}_{t'}, t'))] \quad (4)$$

where $t' = t - \Delta t$ denotes the previous time point, and $d$ is a loss function such as LPIPS, mean squared error, and pseudo-Huber. With the choice of $d(\boldsymbol{x}, \boldsymbol{y}) = \frac{1}{2}\|\boldsymbol{x} - \boldsymbol{y}\|_2^2$, one can derive an alternative objective with equivalent gradients:

$$\min_{\boldsymbol{\theta}} \mathbb{E}_{\boldsymbol{x}, \boldsymbol{\epsilon}, t, \Delta t}[\boldsymbol{f}_{\boldsymbol{\theta}}(\boldsymbol{x}_t, t)^\top (\frac{\Delta \boldsymbol{f}_{\mathrm{sg}[\boldsymbol{\theta}]}(\boldsymbol{x}_t, t)}{\Delta t})], \quad (5)$$

$$\frac{\Delta \boldsymbol{f}_{\mathrm{sg}[\boldsymbol{\theta}]}(\boldsymbol{x}_t, t)}{\Delta t} := \frac{\boldsymbol{f}_{\mathrm{sg}[\boldsymbol{\theta}]}(\boldsymbol{x}_t, t) - \boldsymbol{f}_{\mathrm{sg}[\boldsymbol{\theta}]}(\boldsymbol{x}_s, s)}{\Delta t}. \quad (6)$$

Depending on how we approximate $\boldsymbol{x}_{t-\Delta t}$ given $\boldsymbol{x}_t = \alpha_t \boldsymbol{x} + \sigma_t \boldsymbol{\epsilon}$ in Eq. (6), we obtain consistency distillation (CD)

---

[2]While the original CM objective also contains a time-dependent weight function $w(t)$, we omit it without loss of generality since it can be absorbed into the density function for $t$.

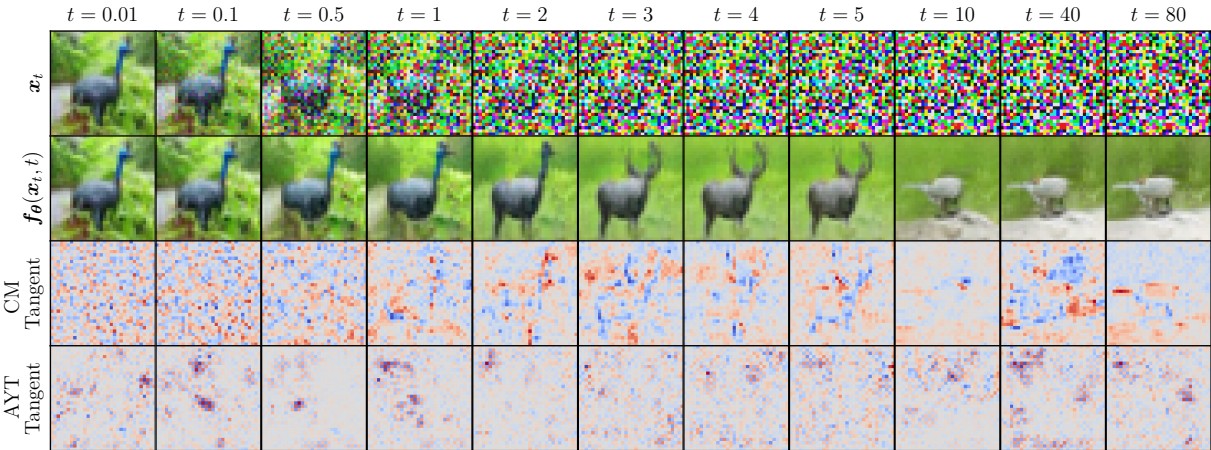

*Figure 2.* CM trajectory tangent visualization on CIFAR10 after training to near-convergence ($400k$ iterations). **First row:** inputs $\boldsymbol{x}_t = \boldsymbol{x}_0 + t\boldsymbol{\epsilon}$. **Second row:** outputs $\boldsymbol{f_\theta}(\boldsymbol{x}_t, t)$. **Third row:** vanilla CM trajectory tangents computed with Eq. (6). Trajectory tangents are averaged along the channel dimension for visualization, and red and blue pixels indicate positive and negative values, resp. **Fourth row:** manifold-aligned trajectory tangents (AYT) computed with Eq. (10).

and consistency training (CT): the former uses $\boldsymbol{x}_{t-\Delta t} \approx \boldsymbol{x}_t - \boldsymbol{v}(\boldsymbol{x}_t, t) \cdot \Delta t$ to distill the velocity, whereas the latter uses $\boldsymbol{x}_{t-\Delta t} \approx \boldsymbol{x}_t - \dot{\boldsymbol{x}}_t \cdot \Delta t \approx \alpha_{t-\Delta t} \boldsymbol{x} + \sigma_{t-\Delta t} \boldsymbol{\epsilon}$.

Letting $\Delta t \to 0$ in Eq. (5) yields the continuous CM objective (Lu & Song, 2025)

$$\min_{\boldsymbol{\theta}} \mathbb{E}_{\boldsymbol{x},\boldsymbol{\epsilon},t}[\boldsymbol{f_\theta}(\boldsymbol{x}_t, t)^\top (\frac{d\boldsymbol{f}_{\mathrm{sg}[\boldsymbol{\theta}]}}{dt}(\boldsymbol{x}_t, t))], \qquad (7)$$

$$\frac{d\boldsymbol{f}_{\mathrm{sg}[\boldsymbol{\theta}]}}{dt}(\boldsymbol{x}_t, t) = \frac{\partial \boldsymbol{f}_{\mathrm{sg}[\boldsymbol{\theta}]}}{\partial \boldsymbol{x}_t}(\boldsymbol{x}_t, t)\frac{d\boldsymbol{x}_t}{dt} + \frac{\partial \boldsymbol{f}_{\mathrm{sg}[\boldsymbol{\theta}]}}{\partial t}(\boldsymbol{x}_t, t). \quad (8)$$

The derivative in Eq. (8) is formally called the *trajectory tangent*, since it is tangential to the trajectory traced out by the CM output $\boldsymbol{f_\theta}(\boldsymbol{x}_t, t)$ as $\boldsymbol{x}_t$ follows the PF-ODE in Eq. (2). The objective possesses a more intuitive formulation

$$\min_{\boldsymbol{\theta}} \mathbb{E}_{\boldsymbol{x},\boldsymbol{\epsilon},t}[\|\boldsymbol{f_\theta}(\boldsymbol{x}_t, t) - \mathrm{sg}[(\boldsymbol{f_\theta}(\boldsymbol{x}_t, t) - \eta d\boldsymbol{f_\theta}/dt)]\|_2^2]$$

as well. Analogous to discrete CMs, we obtain continuous CD or CT depending on how we estimate $d\boldsymbol{x}_t/dt$ in the tangent. CD uses the flow velocity $d\boldsymbol{x}_t/dt = \boldsymbol{v}(\boldsymbol{x}_t, t)$, whereas CT uses $d\boldsymbol{x}_t/dt = \dot{\boldsymbol{x}}_t = \dot{\alpha}_t \boldsymbol{x} + \dot{\sigma}_t \boldsymbol{\epsilon}$.

From Eq. (5) and Eq. (7), we can interpret discrete CM and continuous CM learning from an unified perspective of contracting each path $\{\boldsymbol{f_\theta}(\boldsymbol{x}_s, s) : d\boldsymbol{x}_s = \boldsymbol{v}(\boldsymbol{x}_s, s) ds, \ \boldsymbol{x}_0 = \boldsymbol{x}\}$ along the negative trajectory tangent towards $\boldsymbol{f_\theta}(\boldsymbol{x}_0, 0) = \boldsymbol{x}_0 = \boldsymbol{x}$. The only difference between discrete CM and continuous CM lies in whether we calculate the trajectory tangent using finite differences or the exact derivative. Hence, we may use the trajectory tangent, both discrete and continuous, to analyze training dynamics of CMs.

## 4. Oscillatory Trajectory Tangent Hypothesis

We now further assume data is supported on a low-dimensional manifold $\mathcal{M}$ in $\mathbb{R}^d$ (Narayanan & Mitter, 2010).

Since trajectory tangents represent instantaneous changes in path $\{\boldsymbol{f_\theta}(\boldsymbol{x}_s, s) : d\boldsymbol{x}_s = \boldsymbol{v}(\boldsymbol{x}_s, s) ds, \ \boldsymbol{x}_0 = \boldsymbol{x}\}$, small perturbations of the path can induce large variations in the trajectory tangent. Given the stochasticity within CM training, we hypothesized that trajectory tangents are oscillatory and unlikely to guide the CM output exactly towards the low-dimensional data manifold $\mathcal{M}$. We also hypothesized that this phenomenon actually occurs in practice, and adversely affects CM convergence (see Fig. 1). From here on, these claims will be referred to as the *oscillatory trajectory tangent hypothesis*.

**Observations on CIFAR10.** To validate this hypothesis, we began by examining CM trajectory tangents on CIFAR10 (Krizhevsky, 2009). On CIFAR10, we optimized a CM via consistency training (CT) for $400k$ iterations until near-convergence, so there were no longer large changes in the FID score.[3] We then computed trajectory tangents at various noise levels ranging from $t = 0.01$ to 80. Upon visual inspection of CM trajectory tangents in the third row of Fig. 2, we noticed that *trajectory tangents contained structured patterns that could imply large movements along the manifold, not toward the manifold*, in accordance with our hypothesis.

**Analysis on synthetic data.** To provide further evidence for the oscillatory trajectory tangent hypothesis, we performed an additional experiment on a synthetic dataset with known manifold structure. Specifically, we considered the dataset of images of two-dimensional discs which move vertically or horizontally. As previously noted by Kadkhodaie et al. (2024), this dataset is a two-dimensional curved manifold with tangent space[4] at a point spanned by deformations cor-

---

[3]ECT attains 2-step FID scores of 2.20 at iteration $100k$, and 2.11 at iteration $400k$ (Geng et al., 2025b).

[4]The *tangent space of a manifold* at a point can be intuitively understood as a linear approximation of the manifold at that point

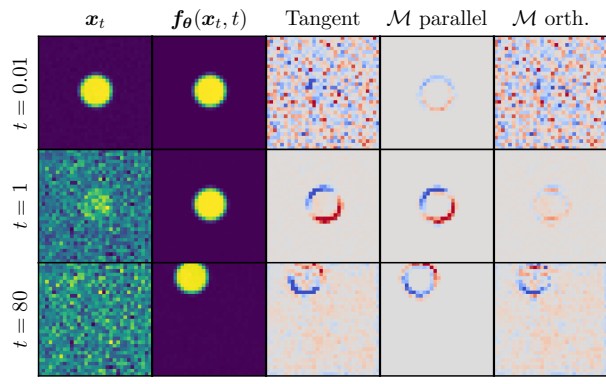

*(a)* Vanilla CM Trajectory Tangent Analysis

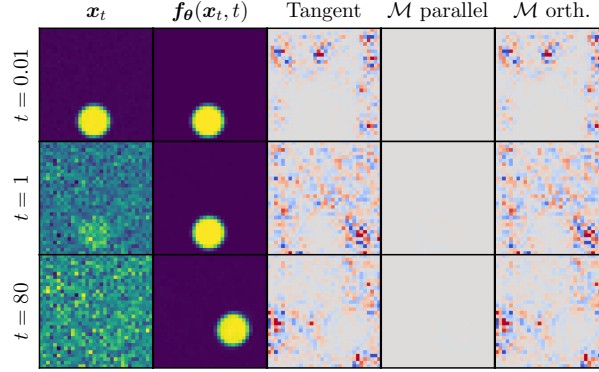

*(b)* AYT (Ours) Trajectory Tangent Analysis

*Figure 3.* Trajectory tangent analysis on 2D discs after training to near-convergence ($200k$ iterations) for vanilla CM and Align Your Trajectory Tangent (AYT). In each figure, we visualize CM inputs, CM outputs, CM trajectory tangents, manifold-parallel component of tangents, and manifold-orthogonal component of tangents.

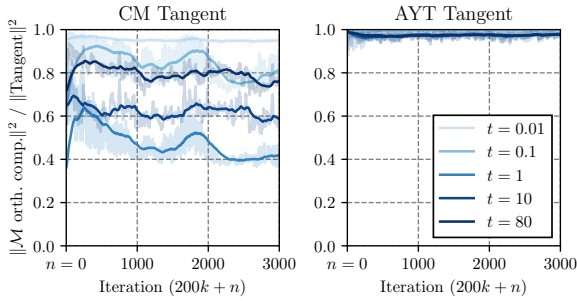

*Figure 4.* Amount of manifold-orthogonal components in trajectory tangents for vanilla CM and our manifold aligned trajectory tangents (AYT) throughout training.

responding to vertical or horizontal movement. Analogous to the previous experiment on CIFAR10, we trained a CM for $200k$ iterations until convergence, and computed tangents at $t \in \{0.01, 0.1, 1, 10, 80\}$ with models at iterations $\geq 200k$.

Motivated by our observations on CIFAR10, we decomposed each trajectory tangent into manifold-parallel and orthogonal components. Concretely, given a CM output $\boldsymbol{f_\theta}(\boldsymbol{x}_t, t)$, we computed its projection onto the manifold $\mathcal{M}$ and the manifold tangent space of $\mathcal{M}$ at the projected point. Let us denote the CM output, its projection, and the manifold tangent space at the projected point as $\boldsymbol{z}$, $\hat{\boldsymbol{z}}$, and $T_{\hat{\boldsymbol{z}}}\mathcal{M}$. Since vectors in $T_{\hat{\boldsymbol{z}}}\mathcal{M}$ lie along $\mathcal{M}$, we defined the manifold-parallel component of the trajectory tangent $d\boldsymbol{f_\theta}(\boldsymbol{x}_t, t)/dt$ as $\mathrm{Proj}_{T_{\hat{\boldsymbol{z}}}\mathcal{M}}(d\boldsymbol{f_\theta}(\boldsymbol{x}_t, t)/dt)$, and the manifold-orthogonal component as the remainder obtained by subtracting the manifold-parallel component from the trajectory tangent. We note that by definition, manifold-parallel and orthogonal components are mutually orthogonal vectors.

Left panel of Fig. 3 displays CM trajectory tangents and their decomposition into manifold-parallel and orthogonal com-

(Lee, 2012), which is distinct from the trajectory tangent.

*Table 1.* Fraction of manifold-orthogonal components in continuous CD trajectory tangents on CIFAR-10 across different conditioning timesteps.

| $t =$ | 0.01 | 0.1 | 1 | 10 | 80 |
|-------|------|-----|---|----|----|
| CM | 0.97 | 0.88 | 0.72 | 0.69 | 0.71 |
| **AYT** | **0.98** | **0.96** | **0.94** | **0.93** | **0.94** |

ponents. Indeed, we see that trajectory tangents are quite oscillatory despite the CM FID having converged – trajectory tangents contain non-trivial manifold-parallel components, especially at $t \geq 1$. This may be concerning, because oscillatory trajectory tangents at large $t$ are at odds with the objective of CM, which is to map pure noise at large $t$ to data. In fact, as shown in the left panel of Fig. 4, we found overwhelmingly large amount of manifold-parallel components in the trajectory tangent. Altogether, there were strong evidences which corroborated the oscillatory trajectory tangent hypothesis. This motivated us to design a loss function which amplifies manifold-relevant components for CM training.

**Validation on CIFAR10.** To further quantify this behavior, we conducted an additional experiment comparing the manifold-orthogonal components of CM and AYT trajectory tangents, following the same protocol as in the synthetic-data analysis. We computed trajectory tangents using continuous-time consistency distillation, which removes the effects of conditional–marginal velocity mismatch and ODE discretization error. Since the CIFAR-10 data manifold is not available in closed form, we approximate it using a CIFAR-10 autoencoder. As shown in Tab. 1, CM exhibits a substantially smaller manifold-orthogonal fraction at larger input timesteps, indicating that a larger portion of its trajectory tangent lies parallel to the data manifold. In contrast, AYT maintains a consistently high manifold-orthogonal fraction across all timesteps. These results suggest that tangent oscillation also arises in a more realistic setting such as

CIFAR-10, and further indicate that model approximation error, beyond conditional–marginal velocity mismatch, is a primary source of manifold-parallel trajectory tangents.

# 5. Align Trajectory Tangents for CM Training

## 5.1. Consistency Model Training with Feature Distance

Kim et al. (2025a) demonstrated that using loss functions of the form $\ell(\boldsymbol{x}, \boldsymbol{y}) = \|\phi(\boldsymbol{x}) - \phi(\boldsymbol{y})\|_2^2$ with an invertible linear map $\phi$ for flow matching can accelerate flow model convergence by amplifying certain directions in the model gradient. For instance, the loss function with $\phi = \boldsymbol{I} + \lambda \operatorname{HPF}$, where HPF is a high-pass filter, magnifies gradient components in the high-frequency regime by a factor of $\lambda + 1$. Taking inspiration from this observation, we adopt a similar approach for designing a new loss function for training consistency models (CMs).

**Proposition 5.1** (Feature-space CM objective equivalence)**.** *Let $\phi : \mathbb{R}^d \to \mathbb{R}^n$ be a (not necessarily linear) feature map and define the feature distance $d_{\phi}(\boldsymbol{x}, \boldsymbol{y}) = \|\phi(\boldsymbol{x}) - \phi(\boldsymbol{y})\|_2$. Consider the consistency matching (CM) loss defined by the squared feature distance. In the continuous-time limit $\Delta t \to 0$, the gradient of the CM objective with respect to $\boldsymbol{\theta}$ is equivalent to that of the objective*

$$\min_{\boldsymbol{\theta}} \ \mathbb{E}_{\boldsymbol{x}, \boldsymbol{\epsilon}, t}\big[\boldsymbol{f}_{\boldsymbol{\theta}}(\boldsymbol{x}_t, t)^{\top} \operatorname{sg}[\boldsymbol{g}_{\phi}(\boldsymbol{x}_t, t)]\big], \quad (9)$$

*where*

$$\boldsymbol{g}_{\phi}(\boldsymbol{x}_t, t) = \left(\frac{d}{dt}\phi(\boldsymbol{f}_{\boldsymbol{\theta}}(\boldsymbol{x}_t, t))\right)^{\top} \boldsymbol{J}_{\phi}(\boldsymbol{f}_{\boldsymbol{\theta}}(\boldsymbol{x}_t, t))$$

$$= \sum_{i=1}^{n} \frac{d}{dt}\phi_i(\boldsymbol{f}_{\boldsymbol{\theta}}(\boldsymbol{x}_t, t)) \nabla_{\boldsymbol{f}_{\boldsymbol{\theta}}}\phi_i(\boldsymbol{f}_{\boldsymbol{\theta}}(\boldsymbol{x}_t, t)) \quad (10)$$

*where $\boldsymbol{J}_{\phi}(\boldsymbol{f}_{\boldsymbol{\theta}}(\boldsymbol{x}_t, t))$ is the Jacobian of $\phi$ w.r.t. $\boldsymbol{f}_{\boldsymbol{\theta}}(\boldsymbol{x}_t, t)$.*

*Proof.* See Appendix C. ∎

It follows that when we use the squared feature distance as the loss, the $d$-dimensional vector Eq. (10) plays the role of CM trajectory tangent during optimization. We observe that Eq. (10) is a linear combination of the rows of the Jacobian of $\phi$, so $\phi$ completely determines which direction the trajectory tangent points to. Thus, with a judiciously chosen $\phi$, one can potentially suppress oscillatory components in the trajectory tangent. However, when $\phi = \operatorname{id}_{\mathbb{R}^d}$, which is the case of the original CM, the Jacobian becomes the full-rank identity matrix $\boldsymbol{I}_d$, so the trajectory tangent is computed as a linear combination of the standard basis, and is free to point in any direction. It turns out that, to align trajectory tangents toward the data manifold, one should use manifold features, which we present in the next section.

## 5.2. Align Trajectory Tangent with Manifold Features

Eq. (10) along with our observations in Section 4 implies that an ideal feature map $\phi$ for optimizing CMs should possess Jacobians whose rows, *i.e.*, gradients $\nabla_{\boldsymbol{z}}\phi_i(\boldsymbol{z})$ for $i = 1, \ldots, n$ point toward the data manifold $\mathcal{M}$. To this end, we consider $\phi$ such that for each coordinate $i$, its level set at zero $\phi_i^{-1}(0) = \mathcal{M}$, and $\phi_i^{-1}(\alpha)$ for increasing values of $|\alpha|$ correspond to increasingly perturbed versions of $\mathcal{M}$.[5] Since the gradient of a scalar-valued function is orthogonal to its level set, we can expect $\nabla_{\boldsymbol{z}}\phi_i(\boldsymbol{z})$ would also point towards $\mathcal{M}$, depending on how the manifold is perturbed. Hence, we shall call each $\phi_i$ a *manifold feature*, and $d_{\phi}$ as a *manifold feature distance*.

In our work, we consider pointwise manifold perturbations of the form $\mathcal{T}_{\alpha}\mathcal{M} \coloneqq \{\mathcal{T}_{\alpha}\boldsymbol{x} : \boldsymbol{x} \in \mathcal{M}\}$, where $\mathcal{T}_{\alpha} : \mathbb{R}^d \to \mathbb{R}^d$ is a transformation smoothly parametrized by $\alpha \in \mathbb{R}$ with $\mathcal{T}_0 = \operatorname{id}_{\mathbb{R}^d}$. Given a collection of $n$ such transformations $\{\mathcal{T}^i\}_{i=1}^{n}$, we can parametrize $\phi$ with a neural net and optimize

$$\min_{\phi} \mathbb{E}_{\boldsymbol{x}, i \in [n], \alpha \in \mathbb{R}}[\|\phi_i(\mathcal{T}_{\alpha}^i(\boldsymbol{x})) - \alpha\|_2^2] \quad (11)$$

such that $\phi_i(\boldsymbol{x}) = \alpha$ for $\boldsymbol{x} \in \mathcal{T}_{\alpha}^i\mathcal{M}$. In particular, with optimal $\phi$, $\phi_i(\boldsymbol{x}) = 0$ for all $\boldsymbol{x} \in \mathcal{M}$ due to the condition $\mathcal{T}_0 = \operatorname{id}_{\mathbb{R}^d}$. We also remark that while isotropic perturbation of $\mathcal{M}$ via, *e.g.*, Gaussian noise addition may be sufficient to generate manifold-orthogonal feature gradients, it can also be beneficial to use anisotropic transformations to further emphasize certain off-manifold directions.

**Transformation and implementation details.** We further limit our scope to the image domain, and consider image transformations for $\mathcal{T}$. We consider three image degradations given by Gaussian noise perturbation, Gaussian blur, and Mixup (Zhang et al., 2017), four geometric transformations given by isotropic scaling, anisotropic scaling, fractional rotation, and fractional translation, and four color transformations given by perturbations in brightness, contrast, hue, and saturation. This yields a feature space of dimension $n = 15$. Thus, components of the trajectory tangent that are amplified by AYT tends to be image degradations, and distinct from directions necessary for diverse generation on the data manifold. This design naturally places greater emphasis on correcting off-manifold deviations while still preserving essential on-manifold information. Readers are referred to Appendix A for a comprehensive description of how the transformations are defined. Manifold feature $\phi$ is parametrized with a VGG16 classification network (Simonyan & Zisserman, 2015), and in the spirit of LPIPS (Zhang et al., 2018), we also use intermediate max-pooling features as manifold features.

---

[5] By $\phi_i^{-1}(\alpha)$, we mean the level set of $\phi_i$ at $\alpha$, *i.e.*, $\phi_i^{-1}(\alpha) \coloneqq \{\boldsymbol{x} \in \mathbb{R}^d : \phi_i(\boldsymbol{x}) = \alpha\}$.

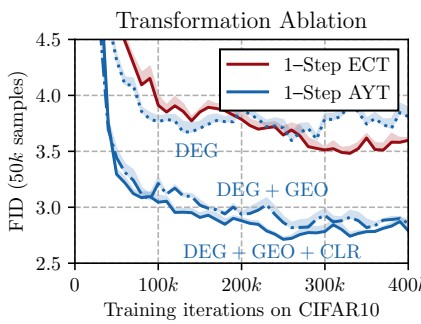 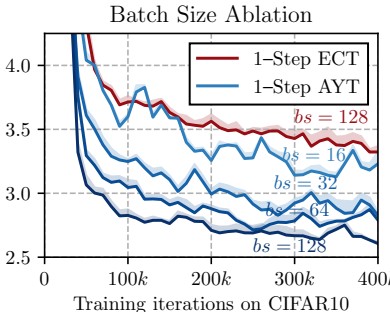 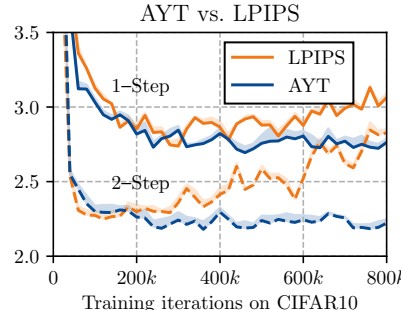

*Figure 5.* Ablation studies on CIFAR10. For transformation ablation and AYT vs. LPIPS, we use batch size 64. Shaded regions indicate min/max FIDs over three generation trials.

*Example for Gaussian Blur.* The transformation is given as $\mathcal{T}_\alpha(\boldsymbol{x}) := \boldsymbol{\kappa}_\alpha \circledast \boldsymbol{x}$, where $\boldsymbol{\kappa}_\alpha$ is a blurring kernel with standard deviation $\alpha$. Manifold feature can be learned by

$$\min_\phi \mathbb{E}_{\boldsymbol{x}, \alpha \sim \text{unif}(0, \alpha_{\max})}[(\phi(\boldsymbol{\kappa}_\alpha \circledast \boldsymbol{x}) - \alpha)^2] \quad (12)$$

where $\text{unif}(0, \alpha_{\max})$ is a uniform distribution on $[0, \alpha_{\max}]$, and discrete CM optimization by

$$\min_{\boldsymbol{\theta}} \mathbb{E}_{\boldsymbol{x}, \boldsymbol{\epsilon}, t, \Delta t}[\frac{1}{\Delta t}(\phi(\boldsymbol{f_\theta}(\boldsymbol{x}_t, t)) - \phi(\boldsymbol{f}_{\text{sg}[\boldsymbol{\theta}]}(\boldsymbol{x}_{t'}, t')))^2]. \quad (13)$$

In other words, in contrast to standard CMs—where $\phi$ is fixed to $\text{id}_{\mathbb{R}^d}$—AYT learns $\phi$ dynamically.

**Conceptual comparison with LPIPS.** Previous works such as (Song et al., 2023) and (Kim et al., 2024a) have used LPIPS for training CMs. Given that LPIPS also uses classifier features to define a distance between images, one may question the novelty of the manifold feature distance. Our distance distinguishes itself from LPIPS in two levels. First, our distance is tailor suited to improving CM training by aligning the trajectory tangent towards the data manifold. Second, the construction of manifold feature distance requires no human supervision and is completely self-supervised, whereas LPIPS requires ImageNet class labels and a human curated dataset of patch similarities. Furthermore, as we shall show in Section 6, CMs trained with manifold feature distance beats CMs trained with LPIPS, and LPIPS suffers from FID degradation possibly due to mismatch between dataset representations.

## 6. Experiments

This section presents various experiments and ablation studies. Additional experiments are provided in Appendix B.

### 6.1. Ablation Studies

**Sanity check in controlled settings.** To verify whether the manifold feature distance suppresses oscillatory components in trajectory tangents, we repeated the experiments in Section 4 with our loss in place of the mean squared error

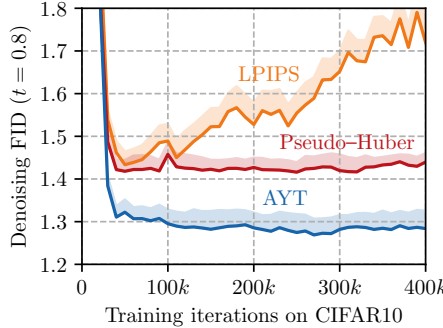

*Figure 6.* Comparison of denoising FIDs at $t = 0.8$ for CMs trained by LPIPS, pseudo-huber (PH), and manifold feature distance (AYT).

(MSE). Concretely, the trajectory tangents were now computed with Eq. (10) instead of Eq. (8). In the bottom row of Fig. 2, the trajectory tangents were scattered and sparse, possibly implying the removal of off-manifold noise. To confirm our intuition, on the two-dimensional discs dataset, we again decomposed trajectory tangents into manifold-parallel and orthogonal components, and computed the amount of manifold-orthogonal component in the trajectory tangent. Manifold feature distance successfully removed oscillatory components from the trajectory tangent, as corroborated by the dominance of manifold-orthogonal components (right panels of Fig. 3 and Fig. 4).

**Transformation ablations.** As mentioned in Section 5.2, we considered three groups of transformations to train manifold features: three degradation-based transformations (DEG), four geometric transformations (GEO), and four color transformations (CLR). The left panel of Fig. 5 shows changes in CM learning curves as $\phi$ was trained with increasing number of transformations. We observed that compounding transformations was always beneficial for CM training. Especially, the addition of geometric transformations led to the largest improvement in FID scores, implying vanilla tangents fail to provide strong training signal towards the data manifold in geometric directions.

**Robustness to batch size.** The middle panel of Fig. 5 displays learning curves when training with batch sizes in

*Table 2.* Sample quality on unconditional CIFAR10 and class-conditional ImageNet $64 \times 64$.

**Unconditional CIFAR10**

| Method | NFE | FID |
|---|---|---|
| **Diffusion models & Fast Samplers** | | |
| DDPM | 1000 | 3.17 |
| EDM (Karras et al., 2022) | 35 | 2.01 |
| DPM-Solver++ (Lu et al., 2025) | 10 | 2.91 |
| DPM-Solver-v3 (Zheng et al., 2023b) | 10 | 2.51 |
| **Diffusion Distillation** | | |
| DFNO (LPIPS) (Zheng et al., 2023a) | 1 | 3.78 |
| PD (Salimans & Ho, 2022) | 1 | 8.34 |
| | 2 | 5.58 |
| TRACT (Berthelot et al., 2023) | 1 | 3.78 |
| | 2 | 3.32 |
| DMD (Yin et al., 2024b) | 1 | 3.77 |
| SiD (Zhou et al., 2024a) | 1 | 1.92 |
| CTM (Kim et al., 2024a) | 1 | **1.87** |
| **Consistency Training** | | |
| iCT (Song & Dhariwal, 2024) | 1 | 2.83 |
| | 2 | 2.46 |
| iCT-deep (Song & Dhariwal, 2024) | 1 | **2.51** |
| | 2 | 2.24 |
| ECT (Geng et al., 2025b) | 1 | 3.60 |
| | 2 | 2.11 |
| ECT+AYT (Ours) | 1 | 2.61 |
| | 2 | 2.13 |

**Class-Conditional ImageNet $64{\times}64$**

| Method | NFE | FID |
|---|---|---|
| **Diffusion models & Fast Samplers** | | |
| EDM (Karras et al., 2022) | 79 | 2.44 |
| EDM2 (Karras et al., 2023) | 63 | 1.33 |
| DPM-Solver (Lu et al., 2022) | 20 | 3.42 |
| **Diffusion Distillation** | | |
| DFNO (LPIPS) (Zheng et al., 2023a) | 1 | 7.83 |
| PD (Salimans & Ho, 2022) | 1 | 10.70 |
| | 2 | 4.70 |
| TRACT (Berthelot et al., 2023) | 1 | 7.43 |
| | 2 | 4.97 |
| DMD (Yin et al., 2024b) | 1 | 2.62 |
| DMD2 (Yin et al., 2024a) | 1 | **1.28** |
| SiD (Zhou et al., 2024a) | 1 | 1.52 |
| CTM (Kim et al., 2024a) | 1 | 1.92 |
| | 2 | 1.73 |
| **Consistency Training** | | |
| iCT (Song & Dhariwal, 2024) | 1 | 4.02 |
| | 2 | 3.20 |
| iCT-deep (Song & Dhariwal, 2024) | 1 | 3.25 |
| | 2 | 2.77 |
| ECT-S (Geng et al., 2025b) | 1 | 5.51 |
| | 2 | 3.18 |
| ECT-S+AYT (Ours) | 1 | 4.42 |
| | 2 | 3.27 |

$\{16, 32, 64, 128\}$. Surprisingly, AYT exhibited strong FID scores even when trained with batch sizes as small as 16, and beat ECT trained with batch size 128. This result further affirms the oscillatory tangent hypothesis, and shows that removing oscillatory components from tangents is crucial for reducing variance during training.

**AYT vs. LPIPS.** In the right panel of Fig. 5, AYT beats LPIPS in both one and two-step generation. In particular, CM trained with LPIPS exhibited severe degradation in FIDs after $400k$ iterations whereas FIDs for AYT showed consistent improvement with more training. We found that this pathology with LPIPS was caused by inaccurate CM outputs at small $t$ corrupting outputs at larger $t$.

As shown in Fig. 6, denoising FIDs at $t = 0.8$ (FID between data $x_0$ and denoised samples $f_\theta(x_0 + t\epsilon, t)$) for CMs trained with LPIPS diverged rapidly after $50k$ steps. We speculate that this behavior arises from the distributional mismatch between ImageNet and CIFAR10 (LPIPS is trained on ImageNet). But, it is unclear how one can generalize LPIPS to other datasets without human supervision, highlighting yet another advantage of our method: AYT presents a simple and interpretable self-supervised pipeline for constructing manifold features on arbitrary datasets, enabling CM training with unbiased representations.

**AYT vs. ECT.** The ablation results in Tab. 3 highlight the

*Table 3.* Comparison of ECT variants with AYT in terms of FID score on CIFAR10. *iters.* denotes number of training steps, AE denotes ECT loss computed with AutoEncoder features, and Adv. denotes ECT combined with adversarial loss.

| | 1 step | 2 steps |
|---|---|---|
| ECT ($400k$ iters.) | 3.48 | 2.13 |
| ECT ($800k$ iters.) | 3.15 | 2.15 |
| ECT + AE ($400k$ iters.) | 3.08 | 2.34 |
| ECT + Adv. ($400k$ iters.) | 3.63 | 2.19 |
| AYT ($400k$ iters.) | 2.71 | 2.17 |

effect of training duration, feature choice, and adversarial supervision on CIFAR-10 FID. Extending ECT training from 400k to 800k iterations improves 1-step FID from 3.48 to 3.15, but provides only marginal benefit and does not close the gap to AYT, suggesting that matching AYT through scaling alone would be inefficient. Replacing ECT features with autoencoder features further improves the 1-step result to 3.08, indicating that feature quality is important, although performance remains below AYT. In contrast, adding adversarial training degrades the final 1-step FID to 3.63, despite potentially accelerating early training, suggesting overfitting or insufficient discriminator strength. Overall, these ablations show that improved feature representations are more effective than longer training or naive adversarial objectives, while AYT remains the strongest 1-step configuration.

## 6.2. Comparison with other methods

We report FID scores across methods and numbers of function evaluations (NFE) in Tab. 2. Additional metrics are reported in Tab. 5 of Appendix B.2.

**Comparison within consistency models.**

On CIFAR10, AYT improves the 1-step FID from 3.60 to 2.61 over Easy Consistency Training (ECT), while maintaining comparable 2-step performance (2.11 vs. 2.13). Notably, AYT also outperforms Improved Consistency Training (iCT) (2.83 FID), despite the latter relying on multi-stage training schedules that progressively reduce timestep gaps. On ImageNet $64 \times 64$, AYT outperforms ECT by a nontrivial margin in both 1- and 2-step settings, reducing 1-step FID from 5.51 to 4.41, and maintains competitive 2-step performance (3.27 vs. 3.18). It also achieves competitive performance relative to iCT, while using significantly fewer resources—most notably, a batch size of 128, which is $8\times$ smaller than the 1024 used by iCT. These results highlight the effectiveness of our tangent alignment strategy in stabilizing consistency model training, without the need for schedule tuning, multi-stage optimization, or large-scale training.

**Comparison with distillation.** On CIFAR10, our method achieves competitive performance compared to SoTA distillation models such as Consistency Trajectory Model (CTM, FID 1.87) and Score Identity Distillation (SiD, FID 1.92), despite not relying on any pretrained teacher model or adversarial training. On Imagenet $64 \times 64$, AYT outperforms several distillation approaches while reducing the gap between SoTA distillation approaches. This result is particularly notable given these baselines often inherit strong priors and score functions from large pretrained diffusion models. In contrast, we train our model from scratch, yet reach comparable or superior sample quality.

**Comparison with fast samplers.** We compare our method with high-order diffusion ODE solvers. Our 2-step performance surpasses that of methods such as DPM-Solver++ and DPM-Solver-v3 that operate with NFE $\geq 10$, despite our significantly smaller sampling cost.

**Comparison on higher resolution.** Finally, we extend our evaluation to a higher-resolution text-to-image setting on the CC12M dataset and compare AYT with Diffusion, Latent Consistency Model (LCM) with Pseudo-Huber loss, and LCM with LPIPS loss. As shown in Tab. 4, AYT consistently outperforms the baselines in terms of both FID and CLIP score across all sampling budgets, including 1-, 2-, and 4-step generation. This demonstrates that our method remains effective beyond low-resolution pixel-space generation, and can be readily applied to latent-space consistency distillation. We refer to Fig 13 for samples.

*Table 4.* Text-to-image consistency distillation results using SD 1.5 on CC12M at $512 \times 512$ resolution under different losses.

| Method | FID ↓ | | | CLIP-Score ↑ | | |
|---|---|---|---|---|---|---|
| | 1 step | 2 steps | 4 steps | 1 step | 2 steps | 4 steps |
| Diffusion | 429.52 | 402.12 | 39.79 | 19.54 | 22.13 | 25.61 |
| Pseudo-Huber | 61.11 | 20.87 | 13.33 | 25.93 | 29.32 | 30.23 |
| LPIPS | 217.84 | 127.96 | 38.59 | 19.54 | 22.12 | 25.65 |
| AYT | **50.61** | **15.39** | **11.53** | **26.31** | **29.65** | **30.34** |

This experiment also highlights another practical advantage of AYT over LPIPS. Since LPIPS is originally defined and trained in pixel space, directly applying it in the latent domain introduces a domain mismatch. To mitigate this issue, we optimized the LPIPS loss after decoding the latent outputs through the VAE; however, this approach still yielded substantially worse performance, as shown in Tab. 4. In contrast, AYT only requires training the manifold feature distance using domain-relevant augmentations, and is therefore agnostic to the domain space of the consistency model. These results suggest that AYT improves over LPIPS not only in sample quality, but also in flexibility across different generative modeling domains.

## 7. Conclusion

In this paper, we analyzed the training dynamics of consistency models (CMs) and showed that their update directions (trajectory tangents) often contain manifold-parallel oscillatory components, which slow convergence. Motivated by this, we introduced the MFD – a simple, self-supervised objective computed in the feature space of an auxiliary network trained to be sensitive to off-manifold perturbations. By aligning trajectory tangents toward the data manifold (i.e., amplifying manifold-orthogonal components), MFD contracts CM trajectories more efficiently without relying on human supervision or curated perceptual datasets. Empirically, MFD stabilizes training by orders of magnitude over the pseudo-Huber loss while improving sample quality. On CIFAR10 and class-conditional ImageNet $64 \times 64$, our method outperforms consistency-training baselines, attains FIDs competitive with distillation approaches despite training from scratch, and remains robust even with very small batch sizes (e.g., 16). These results indicate that matching the optimization geometry of CMs to the data manifold structure is a practical and powerful route to faster, more reliable few-step generation.

## Reproducibility Statement

We describe all experimental procedures in Appendix A, and code for our main experiments as well as checkpoints for feature networks are published at https://github.com/1202kbs/AYT.

## Impact Statement

The goal of this work is to contribute to ongoing advances in machine learning, especially consistency model. Although developments in this area can have broader societal effects, we believe there are no immediate ethical concerns or consequences that warrant special discussion here.

## Acknowledgements

This work was supported by the National Research Foundation of Korea under Grant RS-2024-00336454, the AI Computing Infrastructure Enhancement (GPU Rental Support) User Support Program funded by the Ministry of Science and ICT (MSIT), Republic of Korea (RQT-25-120217), and the Advanced GPU Utilization Support Program funded by the Government of the Republic of Korea (Ministry of Science and ICT) (02-26-01-0404).

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

# A. Experiment Settings

We build our method on top of the Easy Consistency Training (ECT) (Geng et al., 2025b) framework with minor modifications. Unless otherwise noted, we follow ECT defaults for data preprocessing, forward process $x_t = x + t\epsilon$, timestep sampling, and evaluation protocol.

## A.1. Data Preprocessing

CIFAR10 and ImageNet $64 \times 64$ datasets are preprocessed with code provided by (Karras et al., 2022) in `https://github.com/NVlabs/edm`.

## A.2. Model Architectures and Initialization

**Classifier architectures.** We use VGG16 classification networks (Simonyan & Zisserman, 2015) to parametrize manifold features. All VGG16 networks are trained from scratch without any special initialization schemes.

**Consistency model architectures.** We adopt the same backbone choices as ECT. Specifically, we use DDPM++ (Song et al., 2021b) for CIFAR10 and EDM2-S (Karras et al., 2023) for ImageNet $64 \times 64$. On both datasets, we initialize the consistency model with a pretrained diffusion model of the corresponding architecture.

## A.3. Classifier Training Configurations

We use identical training configurations on CIFAR10 and ImageNet $64 \times 64$. Specifically, we use the Adam optimizer (Kingma & Ba, 2015) with learning rate 0.0001 and batch size 512. Each manifold feature is trained for $400k$ iterations to minimize Eq. (11). Our color and geometric transformation pipeline largely follows that described in Appendix B of Karras et al. (2020). We describe the augmentation pipeline for degradations below. Specifically, given $x \sim p(x)$,

- **Gaussian noise.** Sample $\alpha \sim \text{unif}(0, \alpha_{\max})$, $\epsilon \sim \mathcal{N}(0, I)$, return $x + \alpha\epsilon$.

- **Gaussian blur.** Sample $\alpha \sim \text{unif}(0, \alpha_{\max})$ , return $\kappa_\alpha \circledast x$.

- **Mixup.** Sample $\alpha \sim \text{unif}(0, 0.5)$ and another datapoint $y$, return $(1 - \alpha)x + \alpha y$.

Here, $\kappa_\alpha$ is a Gaussian blur kernel with sigma $\alpha$.

## A.4. Consistency Model Training Configurations

We use a global batch size of 128 for all runs, except in the batch-size ablation. Exponential moving average (EMA) is enabled throughout training, with dataset-specific settings detailed below.

**CIFAR10.** We train consistency models for 400K iterations without any multi-stage schedule unlike iCT/ECT. We use the RAdam optimizer (Liu et al., 2020) with learning rate 0.0001 and exponential moving average (EMA) decay rate of 0.9999.

**ImageNet** $64 \times 64$. We train for $200k$ iterations with the same multi-stage schedule as ECT. To mitigate early-stage overfitting, we enable our loss (AYT) after 75K iterations. We use the Adam optimizer (Kingma & Ba, 2015) with learning rate 0.001 and an inverse-square-root decay schedule of decay parameter 2000. EMA follows the Power-EMA formulation introduced in EDM2, but we do not apply post-hoc EMA after training.

**Conceptual 12M.** For the higher-resolution text-to-image experiments on CC12M, we follow the standard LCM-LoRA distillation setup using Stable Diffusion 1.5 as the teacher model. We train at $512 \times 512$ resolution for 1K iterations with a global batch size of 256. The student model is fine-tuned with LoRA of rank 64, while the VAE, text encoder, and teacher U-Net are kept frozen. We use mixed-precision training with fp16, 8-bit AdamW, learning rate $1 \times 10^{-4}$, and zero weight decay. All compared losses, including pseudo-Huber, LPIPS, and AYT, are trained under the same configuration for a fair comparison.

## A.5. Sampling and Evaluation

We evaluate 1-step and 2-step sampling with Fréchet Inception Distance (FID), computed between the training set and 50K generated samples. For 2-step sampling, the intermediate timestep is fixed to $t = 0.821$ on CIFAR10 and $t = 1.526$ on

*Table 5.* Additional metrics comparing consistency models.

| Method | NFE ($\downarrow$) | FID ($\downarrow$) | IS ($\uparrow$) | Precision ($\uparrow$) | Recall ($\uparrow$) | Density ($\uparrow$) | Coverage ($\uparrow$) |
|---|---|---|---|---|---|---|---|
| **CIFAR10** | | | | | | | |
| CT (LPIPS) (Song et al., 2023) | 1 | 8.70 | 8.49 | - | - | - | - |
| | 2 | 5.83 | 8.85 | - | - | - | - |
| CD (LPIPS) (Song et al., 2023) | 1 | 3.55 | 9.48 | - | - | - | - |
| | 2 | 2.93 | 9.75 | - | - | - | - |
| iCT (Song & Dhariwal, 2024) | 1 | 2.83 | 9.54 | - | - | - | - |
| | 2 | 2.46 | 9.80 | - | - | - | - |
| ECT (Geng et al., 2025b) | 1 | 3.60 | 9.80 | 0.69 | 0.75 | 0.79 | 0.92 |
| | 2 | 2.11 | 10.09 | 0.72 | 0.75 | 0.88 | 0.94 |
| ECT+AYT (Ours) | 1 | 2.61 | 9.98 | 0.71 | 0.75 | 0.85 | 0.93 |
| | 2 | 2.13 | 10.16 | 0.72 | 0.74 | 0.88 | 0.94 |
| **ImageNet** $64 \times 64$ | | | | | | | |
| CT (LPIPS) (Song et al., 2023) | 1 | 13.0 | - | 0.71 | 0.47 | - | - |
| | 2 | 11.1 | - | 0.69 | 0.56 | - | - |
| CD (LPIPS) (Song et al., 2023) | 1 | 6.20 | - | 0.68 | 0.63 | - | - |
| | 2 | 4.70 | - | 0.69 | 0.64 | - | - |
| iCT (Song & Dhariwal, 2024) | 1 | 4.02 | - | 0.70 | 0.63 | - | - |
| | 2 | 3.20 | - | 0.73 | 0.63 | - | - |
| ECT-S (Geng et al., 2025b) | 1 | 5.51 | 17.17 | 0.78 | 0.46 | 0.92 | 0.51 |
| | 2 | 3.18 | 21.35 | 0.77 | 0.49 | 0.88 | 0.55 |
| ECT-S+AYT (Ours) | 1 | 4.42 | 19.00 | 0.78 | 0.47 | 0.90 | 0.52 |
| | 2 | 3.27 | 22.22 | 0.77 | 0.50 | 0.89 | 0.55 |
| **ImageNet** $512 \times 512$ | | | | | | | |
| ECT (Geng et al., 2025b) | 1 | 27.479 | 27.55 | 0.77 | 0.58 | 0.63 | 0.89 |
| ECT+AYT (Ours) | 1 | 27.029 | 27.85 | 0.77 | 0.58 | 0.63 | 0.88 |

ImageNet $64 \times 64$, following ECT. Unless otherwise stated, we follow the ECT evaluation setup and report FID computed with $50k$ samples.

# B. Additional Experiments

## B.1. Preliminary Results on ImageNet $512 \times 512$

In this section, we compare the generative performance of our method against the baseline ECT on ImageNet $512 \times 512$. We employ SD-VAE (Rombach et al., 2022) as the latent autoencoder. Both ECT and our method are trained for 50k iterations with a batch size of 128, starting from the same pretrained EDM2 checkpoint. The implementation details largely follow those used in the ImageNet $64 \times 64$ experiments, except that we do not adopt the multi-stage schedule. The timestep sampling distribution is identical to that used for EDM2 on ImageNet $512 \times 512$. As shown in Table 5, our method also achieves improved generative performance over ECT in the LDM setting.

In Figure 11, we also provide samples on ImageNet $512 \times 512$. While these are only preliminary results, we observe that AYT samples are less blurry with more recognizable objects.

## B.2. Additional Metrics

In Table 5, we report additional metrics such as the Inception Score (Salimans et al., 2016), precision and recall (Kynkäänniemi et al., 2019), and density and coverage (Naeem et al., 2020). Compared to ECT, AYT showed consistent improvements in quality metrics such as FID, IS, precision, or density. Moreover, the improvement in quality did not come at the cost of sample diversity, as corroborated by competitive recall and coverage scores.

## B.3. Variance Reduction Effects of AYT

The gradient variance measurements in Table 7 support our hypothesis that AYT converges faster than CM by inducing lower-variance stochastic gradients. Since the convergence bound scales with $M_\phi = (\nabla_\theta f_\theta)^\top \text{Cov}(g_\phi)(\nabla_\theta f_\theta)$, smaller

*Table 6.* Manifold feature $\phi$ architecture and training setting ablation on CIFAR10. Wall-clock time is measured on a single RTX 4090. We report 1 and 2-step FIDs for ECT trained for $400k$ iterations with corresponding $\phi$ and batch size 64. **Acceleration** denotes overall speed-up in wall-clock time to reach 1-step FID of baseline ECT, *i.e.*, (time required to train ECT for $400k$ iterations) / (($\phi$ train time) + (time required by AYT to reach 1-step FID of 3.48)).

| $\phi$ Architecture | $\phi$ Batch Size | $\phi$ Train Time | CM Train Time | 1-Step FID | 2-Step FID | Acceleration |
|---|---|---|---|---|---|---|
| VGG16 | 512 | 10 hours | 33 hours | 2.71 | 2.17 | ×2.41 |
| *halve batch size* | 256 | 5 hours | 33 hours | 2.74 | 2.23 | ×3.86 |
| *w/o maxpool* | 256 | 5 hours | 33 hours | 2.73 | 2.19 | ×3.86 |
| *w/o maxpool, batchnorm* | 256 | 5 hours | 33 hours | 2.85 | 2.19 | ×3.86 |
| VGG11 | 256 | 4.5 hours | 33 hours | 2.82 | 2.07 | ×3.71 |
| ResNet | 256 | 5 hours | 33 hours | 2.94 | 2.52 | ×3.22 |
| DenseNet | 256 | 7.5 hours | 33 hours | 2.87 | 2.45 | ×3.21 |
| **Baseline ECT with $\phi = $ identity** | | 0 hours | 32 hours | 3.48 | 2.13 | ×1.0 |

*Table 7.* Gradient variance comparison between CM and AYT. CM corresponds to the identity trajectory tangent $g_I$, while AYT uses the trajectory tangent $g_\phi$. Lower values indicate smaller stochastic gradient variance.

| | CIFAR-10 | | | | | |
|---|---|---|---|---|---|---|
| | enc32 | enc16 | enc8 | dec8 | dec16 | dec32 |
| $M_I$ | $2.3\times10^{-5}$ | $1.1\times10^{-5}$ | $7.3\times10^{-6}$ | $1.3\times10^{-7}$ | $4.0\times10^{-7}$ | $5.1\times10^{-8}$ |
| $M_\phi$ | $1.2\times10^{-6}$ | $6.4\times10^{-7}$ | $3.3\times10^{-7}$ | $5.2\times10^{-9}$ | $6.7\times10^{-9}$ | $1.5\times10^{-9}$ |

| | LCM, $\sigma_D = 1.5$ | | | | | |
|---|---|---|---|---|---|---|
| | down0 | down1 | down2 | up0 | up1 | up2 |
| $M_I$ | $9.95\times10^{-1}$ | $2.47\times10^{-1}$ | $2.10\times10^{-1}$ | $5.63\times10^{-3}$ | $8.30\times10^{-2}$ | $6.31\times10^{-2}$ |
| $M_\phi$ | $2.79\times10^{-1}$ | $1.64\times10^{-1}$ | $1.43\times10^{-1}$ | $4.29\times10^{-3}$ | $6.41\times10^{-2}$ | $4.63\times10^{-2}$ |

values indicate a more favorable optimization geometry. Across both CIFAR-10 and LCM with $\sigma_D = 1.5$, AYT consistently yields lower variance than CM at every measured network block. On CIFAR-10, the reduction is particularly pronounced, with $M_\phi$ often one to two orders of magnitude smaller than $M_I$. The same trend also holds for LCM, where AYT reduces gradient variance across both downsampling and upsampling blocks. These results suggest that the oscillatory CM trajectory tangents $g_I$ induce a more diffuse covariance structure, whereas the manifold-orthogonal AYT tangents $g_\phi$ yield a more concentrated covariance, leading to lower-gradient variance and faster optimization.

### B.4. Wall-Clock Time Comparison

Table 6 shows an ablation of $\phi$ architectures. Starting with VGG16 as $\phi$, halving the batch size for $\phi$ training from 512 to 256 led to a small degradation in FID scores while significantly improving wall-clock acceleration from ×2.41 to ×3.86.

Simplifying VGG16 by removing max-pooling did not hurt performance, but further removing batch normalization led to a slight degradation in one-step FID. Reducing the number of layers by converting to VGG11 led to both losses in 1-step FID and wall-clock acceleration.

We also tested other architectures such as ResNet (He et al., 2016) or DenseNet (Huang et al., 2017), neither of which performed better than VGG. Overall, our observations imply simple architectures work best as manifold features, and one should try reducing batch size first when aiming for compute efficiency.

Here, we provide further detail for the computation of wall-clock acceleration. With batch size 64, Easy Consistency Training (ECT) (Geng et al., 2025b) on CIFAR10 (Krizhevsky, 2009) for $400k$ iterations takes

- 32 hours with Pseudo-Huber loss,

- 33 hours with feature-based losses such as LPIPS or AYT.

Overall, training with feature-based metrics such as LPIPS (Zhang et al., 2018) or manifold feature distance is approximately 3% slower than training with Pseudo-Huber.

To compute wall-clock acceleration for a particular choice of $\phi$, we measured both wall-clock training time and number of training iterations required to reach the final performance of baseline ECT at $400k$ iterations. For instance, with $\phi$ as the VGG16 network trained with batch size 512, $\phi$ optimization consumes 10 hours, and AYT with this $\phi$ surpasses the

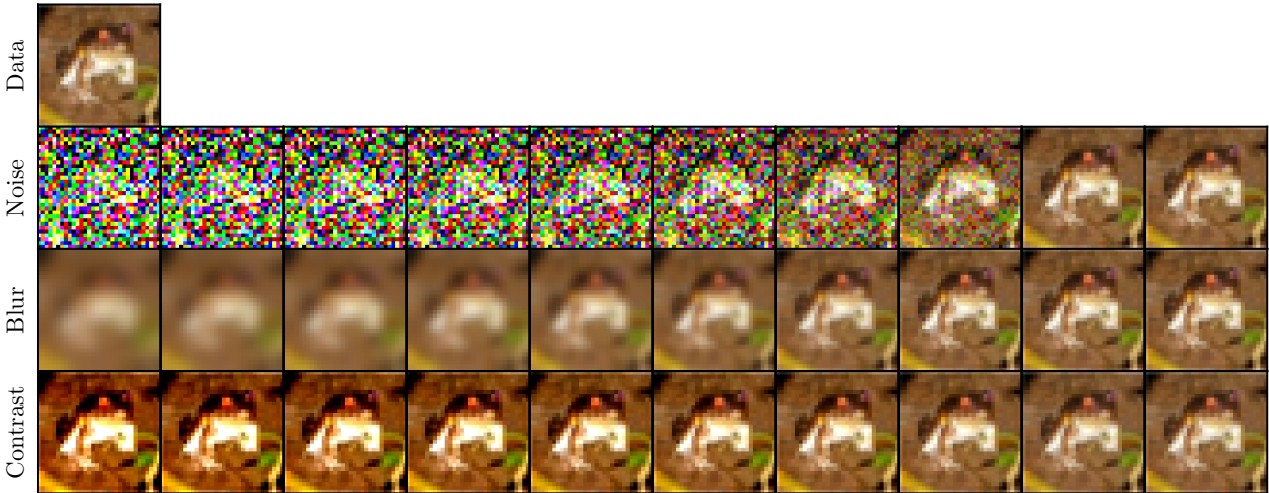

*Figure 7.* Visualization of manifold feature inversion via gradient descent on the objective Eq. (17) with $\beta = 0$ (progress from left to right). Three transformations Gaussian noise adition, Gaussian blur, and contrast are considered. Ground-truth data is shown at the upper left corner.

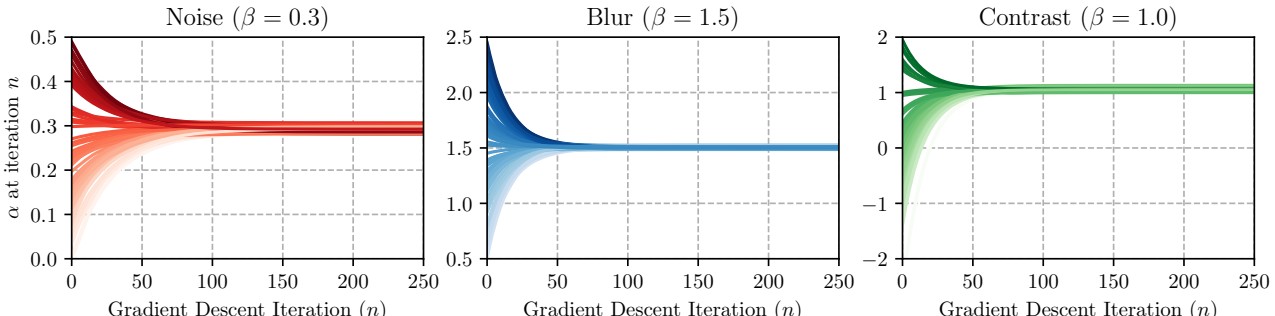

*Figure 8.* Manifold feature inversion for three transformations: Gaussian noise addition, Gaussian blur, and contrast. Transformation parameter $\alpha$ is initialized randomly, and optimized via gradient descent on the objective Eq. (17). Target $\beta$ is written at top of respective panels.

performance of baseline at $40k$ iterations. This implies that the total amount of wall-clock time needed to surpass ECT is

$$10 + 33 \times (40k/400k) = 13.3 \tag{14}$$

so AYT in this case is $32/13.3 \approx 2.41$ times faster than ECT. Wall-clock acceleration for other configurations is calculated in an analogous manner.

### B.5. Manifold Feature Verification via Inversion

In this section, we verify whether the level sets of learned manifold feature $\phi$ capture perturbed manifolds. One way to do so would be to sample an arbitrary point $\boldsymbol{x} \in \mathbb{R}^d$, solve

$$\min_{\boldsymbol{x} \in \mathbb{R}^d} \|\phi(\boldsymbol{x}) - \beta\|_2^2 \tag{15}$$

via gradient descent on $\boldsymbol{x}$, and check whether $\boldsymbol{x} \in \mathcal{T}_\beta \mathcal{M}$ at optimality. However, there are two reasons why this approach is unfeasible in practice.

First, we have no way of verifying whether the optimized $\boldsymbol{x}$ belongs to $\mathcal{T}_\beta \mathcal{M}$ as we do not know the ground-truth data manifold $\mathcal{M}$. Second, because $\cup_\alpha \mathcal{T}_\alpha \mathcal{M} \subsetneq \mathbb{R}^d$ in general, any $\boldsymbol{y} \in \mathbb{R}^d - \cup_\alpha \mathcal{T}_\alpha \mathcal{M}$ will be an out-of-distribution point to $\phi$ because it was not seen by $\phi$ during training. Thus, $\phi$ may not provide meaningful gradients on $\mathbb{R}^d - \cup_\alpha \mathcal{T}_\alpha \mathcal{M}$. So, if $\boldsymbol{x}$ happens to leave $\cup_\alpha \mathcal{T}_\alpha \mathcal{M}$ during gradient descent, it may get lost in $\mathbb{R}^d - \cup_\alpha \mathcal{T}_\alpha \mathcal{M}$ and diverge or converge to a meaningless stationary point.

Hence, instead of solving Eq. (15), we solve

$$\min_{\boldsymbol{y}} \|\phi(\boldsymbol{y}) - \beta\|_2^2 \quad s.t. \quad \boldsymbol{y} \in \cup_\alpha \mathcal{T}_\alpha \boldsymbol{x} \tag{16}$$

or the equivalent problem

$$\min_\alpha \|\phi(\mathcal{T}_\alpha(\boldsymbol{x})) - \beta\|_2^2 \tag{17}$$

and check whether $\alpha = \beta$ is satisfied at optimality. Essentially, this alternative problem constrains the input to $\phi$ to stay within $\cup_\alpha \mathcal{T}_\alpha \mathcal{M}$ and so ensures stable optimization.

In Fig. 7, we visualize gradient descent trajectories on Eq. (17) with initial point $\boldsymbol{y} = \mathcal{T}_\alpha \boldsymbol{x}$ and target $\beta = 0$. Since we use transformations such that $\mathcal{T}_0 = \mathrm{id}_{\mathbb{R}^d}$, optimizing Eq. (17) with $\beta = 0$ should yield $\boldsymbol{y} = \mathcal{T}_0 \boldsymbol{x} = \boldsymbol{x}$ at optimality. We see that it is indeed the case – we gradually recover the initial data point with increasing number of gradient descent steps.

In Fig. 8, we also check whether $\phi$ learns perturbed manifolds by solving Eq. (17) with $\beta \neq 0$. Gradient descent on the objective Eq. (17) results in $\alpha$ values which converge to the target $\beta$ values. This implies that level sets of manifold features $\phi$ properly approximate perturbed data manifolds, and justifies our use of manifold features to compute CM losses with the goal of aligning trajectory tangents towards the data manifold.

## C. Proof of Proposition 5.1

**Proposition C.1** (Feature-space CM objective equivalence). *Let $\phi : \mathbb{R}^d \to \mathbb{R}^n$ be a (not necessarily linear) feature map and define the feature distance $d_\phi(\boldsymbol{x}, \boldsymbol{y}) = \|\phi(\boldsymbol{x}) - \phi(\boldsymbol{y})\|_2$. Consider the consistency matching (CM) loss defined by the squared feature distance. In the continuous-time limit $\Delta t \to 0$, the gradient of the CM objective with respect to $\boldsymbol{\theta}$ is equivalent to that of the objective*

$$\min_{\boldsymbol{\theta}} \ \mathbb{E}_{\boldsymbol{x},\boldsymbol{\epsilon},t}\big[\boldsymbol{f}_{\boldsymbol{\theta}}(\boldsymbol{x}_t, t)^\top \mathrm{sg}[\boldsymbol{g}_\phi(\boldsymbol{x}_t, t)]\big], \tag{18}$$

*where*

$$\boldsymbol{g}_\phi(\boldsymbol{x}_t, t) = \left(\frac{d}{dt}\phi(\boldsymbol{f}_{\boldsymbol{\theta}}(\boldsymbol{x}_t, t))\right)^\top \boldsymbol{J}_\phi(\boldsymbol{f}_{\boldsymbol{\theta}}(\boldsymbol{x}_t, t)). \tag{19}$$

*Equivalently,*

$$\boldsymbol{g}_\phi(\boldsymbol{x}_t, t) = \sum_{i=1}^n \frac{d}{dt}\phi_i(\boldsymbol{f}_{\boldsymbol{\theta}}(\boldsymbol{x}_t, t)) \nabla_{\boldsymbol{f}_{\boldsymbol{\theta}}} \phi_i(\boldsymbol{f}_{\boldsymbol{\theta}}(\boldsymbol{x}_t, t)). \tag{20}$$

*where $\boldsymbol{J}_\phi(\boldsymbol{f}_{\boldsymbol{\theta}}(\boldsymbol{x}_t, t))$ is the Jacobian of $\phi$ w.r.t. $\boldsymbol{f}_{\boldsymbol{\theta}}(\boldsymbol{x}_t, t)$.*

*Proof.* We consider the squared feature-space CM loss

$$\frac{1}{2\Delta t} \left\|\phi(\boldsymbol{f}_{\boldsymbol{\theta}}(\boldsymbol{x}_t, t)) - \phi(\boldsymbol{f}_{\mathrm{sg}[\boldsymbol{\theta}]}(\boldsymbol{x}_{t-\Delta t}, t - \Delta t))\right\|_2^2.$$

Taking the gradient with respect to $\boldsymbol{\theta}$ and using the stop-gradient operator, only the first term contributes. Applying the chain rule yields

$$\frac{1}{\Delta t}\big(\phi(\boldsymbol{f}_{\boldsymbol{\theta}}(\boldsymbol{x}_t, t)) - \phi(\boldsymbol{f}_{\boldsymbol{\theta}}(\boldsymbol{x}_{t-\Delta t}, t - \Delta t))\big)^\top \boldsymbol{J}_\phi(\boldsymbol{f}_{\boldsymbol{\theta}}(\boldsymbol{x}_t, t))\nabla_{\boldsymbol{\theta}}\boldsymbol{f}_{\boldsymbol{\theta}}(\boldsymbol{x}_t, t).$$

Taking the limit $\Delta t \to 0$ gives

$$\left(\frac{d}{dt}\phi(\boldsymbol{f}_{\boldsymbol{\theta}}(\boldsymbol{x}_t, t))\right)^\top \boldsymbol{J}_\phi(\boldsymbol{f}_{\boldsymbol{\theta}}(\boldsymbol{x}_t, t))\nabla_{\boldsymbol{\theta}}\boldsymbol{f}_{\boldsymbol{\theta}}(\boldsymbol{x}_t, t),$$

which coincides with the gradient of the stated objective, completing the proof. $\square$

# D. Proof of Manifold-Orthogonality of Manifold Feature Gradients

**Assumption D.1.** Transformation $\mathcal{T}_\alpha$ is a diffeomorphism whose Jacobian $\boldsymbol{J}_{\mathcal{T}_\alpha}(\boldsymbol{x})$ for each $\boldsymbol{x} \in \mathcal{M}$ possesses $\boldsymbol{v} \in T_{\boldsymbol{x}}\mathcal{M}$ as eigenvectors.

**Assumption D.2.** The level sets of $\phi$ capture perturbed manifolds, *i.e.*, $\phi^{-1}(\alpha) = \mathcal{T}_\alpha \mathcal{M}$.

**Lemma D.3.** *Given a differentiable scalar-valued function $f : \mathbb{R}^d \to \mathbb{R}$, its gradient $\nabla f$ at $\boldsymbol{x} \in \mathbb{R}^d$ is perpendicular to its level set $f^{-1}(\alpha)$, where $\alpha = f(\boldsymbol{x})$.*

*Proof.* Let $\boldsymbol{v}$ be a tangential vector to $f^{-1}(\alpha)$ at $\boldsymbol{x}$. We can construct a curve $\boldsymbol{r}_t$ with $\boldsymbol{r}_0 = \boldsymbol{x}$ and $\dot{\boldsymbol{r}}_0 = \boldsymbol{v}$, where by $\dot{\boldsymbol{r}}_0$, we denote the time-derivative of $\boldsymbol{r}_t$ at $t = 0$ (for a proof of this fact, see Proposition 3.23 in (Lee, 2012)). Because $f(\boldsymbol{r}_t)$ is constant for all $t$,

$$0 = \frac{df(\boldsymbol{r}_t)}{dt}\bigg|_{t=0} = \frac{df(\boldsymbol{r}_t)}{d\boldsymbol{r}_t}\frac{d\boldsymbol{r}_t}{dt}\bigg|_{t=0} = \nabla f(\boldsymbol{r}_0)^\top \dot{\boldsymbol{r}}_0 = \nabla f(\boldsymbol{x})^\top \boldsymbol{v} \tag{21}$$

which proves the Lemma. $\square$

**Proposition D.4.** *Under assumptions D.1 and D.2, manifold feature gradient $\nabla \phi$ is manifold-orthogonal. More rigorously speaking, given $\boldsymbol{x} \in \mathcal{M}$ and $\boldsymbol{y} = \mathcal{T}_\alpha \boldsymbol{x}$,*

$$\nabla \phi(\boldsymbol{y})^\top \boldsymbol{v} = 0 \tag{22}$$

*for all $\boldsymbol{v} \in T_{\boldsymbol{x}}\mathcal{M}$.*

*Proof.* By Lemma D.3, we see that

$$\nabla \phi(\boldsymbol{y})^\top \boldsymbol{w} = 0 \tag{23}$$

for all vectors $\boldsymbol{w}$ tangential to the level set $\phi^{-1}(\alpha)$, or in other words, $\boldsymbol{w} \in T_{\boldsymbol{y}}\phi^{-1}(\alpha) = T_{\boldsymbol{y}}\mathcal{T}_\alpha \mathcal{M}$. Because $\mathcal{T}_\alpha$ is a diffeomorphism, Proposition 3.6 (d) of (Lee, 2012) implies that

$$T_{\boldsymbol{y}}\mathcal{T}_\alpha \mathcal{M} = \boldsymbol{J}_{\mathcal{T}_\alpha}(\boldsymbol{x})T_{\boldsymbol{x}}\mathcal{M} \tag{24}$$

where we view the Jacobian $\boldsymbol{J}_{\mathcal{T}_\alpha}(\boldsymbol{x})$ as a linear transformation on the vectors in $T_{\boldsymbol{x}}\mathcal{M}$. Furthermore, because $\mathcal{T}_\alpha$ is a diffeomorphism, $\boldsymbol{J}_{\mathcal{T}_\alpha}(\boldsymbol{x})$ is a vector space isomorphism from $T_{\boldsymbol{x}}\mathcal{M}$ to $T_{\boldsymbol{y}}\mathcal{T}_\alpha \mathcal{M}$. Hence, for each $\boldsymbol{v} \in T_{\boldsymbol{x}}\mathcal{M}$, there exists $\boldsymbol{w} \in T_{\boldsymbol{y}}\mathcal{T}_\alpha \mathcal{M}$ such that

$$\boldsymbol{w} = \boldsymbol{J}_{\mathcal{T}_\alpha}(\boldsymbol{x})\boldsymbol{v} = \lambda \boldsymbol{v} \tag{25}$$

where $\lambda \neq 0$ is the eigenvalue of $\boldsymbol{J}_{\mathcal{T}_\alpha}(\boldsymbol{x})$ for $\boldsymbol{v}$. Plugging this relation into Eq. (24), we obtain

$$0 = \nabla \phi(\boldsymbol{y})^\top \boldsymbol{w} = \lambda \nabla \phi(\boldsymbol{y})^\top \boldsymbol{v} \tag{26}$$

which implies $\nabla \phi(\boldsymbol{y})^\top \boldsymbol{v} = 0$ because $\lambda$ is nonzero. $\square$

**Corollary D.5.** *Additive transformations such as Gaussian noise addition or Mixup yields manifold-orthogonal manifold feature gradients.*

*Proof.* We observe that additive transformations can be expressed as

$$\mathcal{T}_\alpha(\boldsymbol{x}) = \boldsymbol{x} + \alpha \boldsymbol{\epsilon} \tag{27}$$

where $\boldsymbol{\epsilon}$ is the added perturbation. Then $\boldsymbol{J}_{\mathcal{T}_\alpha}(\boldsymbol{x}) = \boldsymbol{I}$ which trivially satisfies Assumption D.1. $\square$

**Remark.** Most transformations considered in our paper such as additive perturbations, geometric transformations, and color transformations are diffeomorphisms, so the first part of Assumption D.1 is satisfied. While we cannot ascertain whether the eigenvector condition in Assumption D.1 is satisfied as we do not have explicit knowledge of the manifold structure of real-world datasets, the eigenvector condition can intuitively be understood as a requirement that local changes in $\boldsymbol{x} \in \mathcal{M}$ along the data manifold is reflected in transformed outputs $\mathcal{T}_\alpha \boldsymbol{x}$ as well. We believe this is a realistic assumption which would hold for many of our considered transformations.

# E. Discussion

## E.1. Further Implications

Beyond images, it will be interesting to explore other domains such as audio, text, or multimodal data with diverse augmentation strategies. For example, in audio data, common augmentations include time-stretching, pitch-shifting, masking, or adding background noise, all of which can be utilized for learning data manifold features. Applying our approach in such settings could provide valuable insights into how well the proposed distance metric generalizes across modalities. Such extensions could further demonstrate the generality of the framework.

## E.2. Limitations

Our study has so far focused on relatively small-scale settings. While the method requires additional training and increases memory usage, the auxiliary classifier is lightweight: it trains much faster than the main model and adds little memory overhead. As a result, we expect these constraints to be less critical in practice, even when scaling to larger datasets.

We have also restricted our evaluation up to resolution $64 \times 64$. While higher-resolution experiments remain open, the consistent improvements on CIFAR10 and ImageNet suggest that the approach may transfer well to more demanding settings. Moreover, recent high-resolution training often relies on latent diffusion models (LDMs) (Rombach et al., 2022), which downsample images by a factor of 8, making our method potentially well-suited for such pipelines. In this sense, the present work should be viewed as a first step: AYT establishes strong evidence on standard benchmarks while opening several promising directions for scaling and broader applications in self-supervised and generative learning.

## E.3. Further Comparison to Related Works

Here, we provide further comparison to previous works on using representations to improve diffusion or flow training

**Comparison to representation alignment.** Motivations behind representation alignment type of works such as (Yu et al., 2025; Leng et al., 2025) and our work are fundamentally different. The former aims to enhance generative quality of diffusion and flow models by aligning the internal representations of diffusion and flow models to high-quality self-supervised representations. As the authors of REPA wrote in their paper (see Appendix M of (Yu et al., 2025)), there is no solid theoretical intuition as to why representation alignment helps flow and diffusion training. On the other hand, our work pinpoints a source of slow convergence in CMs (Section 4), namely oscillatory tangents, and derives a theoretically principled loss for mitigating them (Section 5).

**Comparison to previous perceptual losses.** The most relevant works to ours are (Lin & Yang, 2024) and (Berrada et al., 2025). Research on perceptual objectives for Consistency Models (CMs) has been limited since the original CM paper. Following the introduction of the pseudo-Huber loss in (Song & Dhariwal, 2024) as an alternative to LPIPS, subsequent CM studies have largely continued to employ pseudo-Huber loss or closely related variants without significant changes. Although perceptual objectives have been shown to improve generative quality in diffusion models—for example, in the aforementioned works—they have been underexplored in the context of CMs despite their potential benefits.

**Comparison to previous consistency model works.** Our contribution is twofold: we revisit perceptual objectives for CMs and improve upon LPIPS-based formulations, and we also identify the previously unaddressed issue of tangent oscillation during training. Existing CM research has primarily focused on reducing error accumulation—for example, by modifying model architecture (Lu & Song, 2025), minimizing discretization errors via JVP techniques (Lu & Song, 2025; Geng et al., 2025a), leveraging pretrained diffusion models or performing additional stages for stabilized initialization (Lu & Song, 2025; Geng et al., 2025b; Hu et al., 2025), or introducing curriculum learning strategies (Zhang et al., 2025).

These approaches all aim to reduce accumulated error across timesteps, but they do not account for the oscillatory behavior we observe in manifold-parallel tangent components. By uncovering this distinct source of slow convergence, we provide a complementary perspective and a novel direction for improving CM training stability. A related line of work is (Issenhuth et al., 2025), but unlike that work, our method focuses more directly on the geometric structure of the data manifold.

## F. Consistency Model Samples

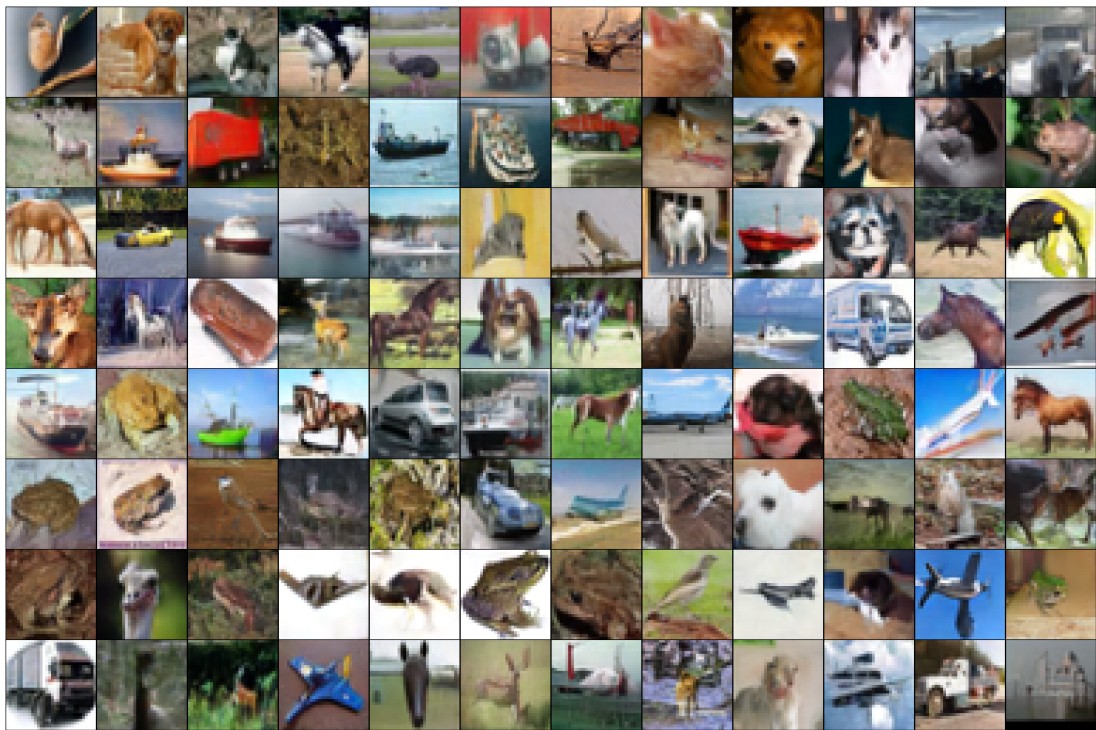

*(a)* Samples from CM trained via ECT.

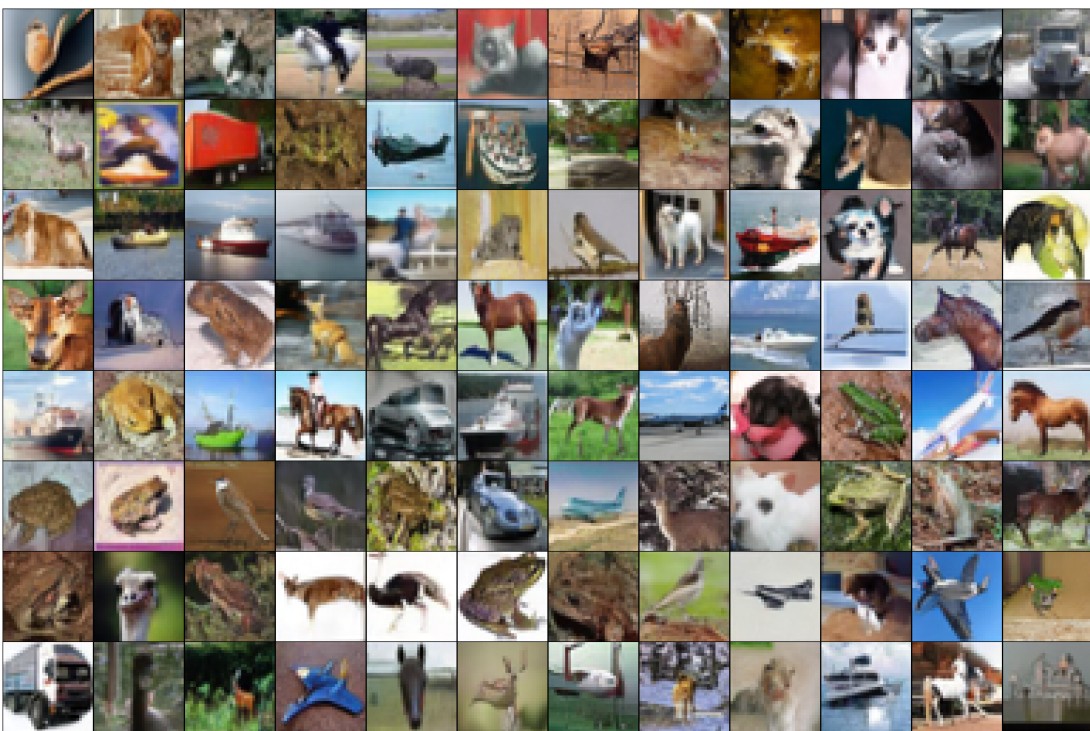

*(b)* Samples from CM trained via AYT (Ours).

*Figure 9.* Uncurated one-step CM samples on CIFAR10.

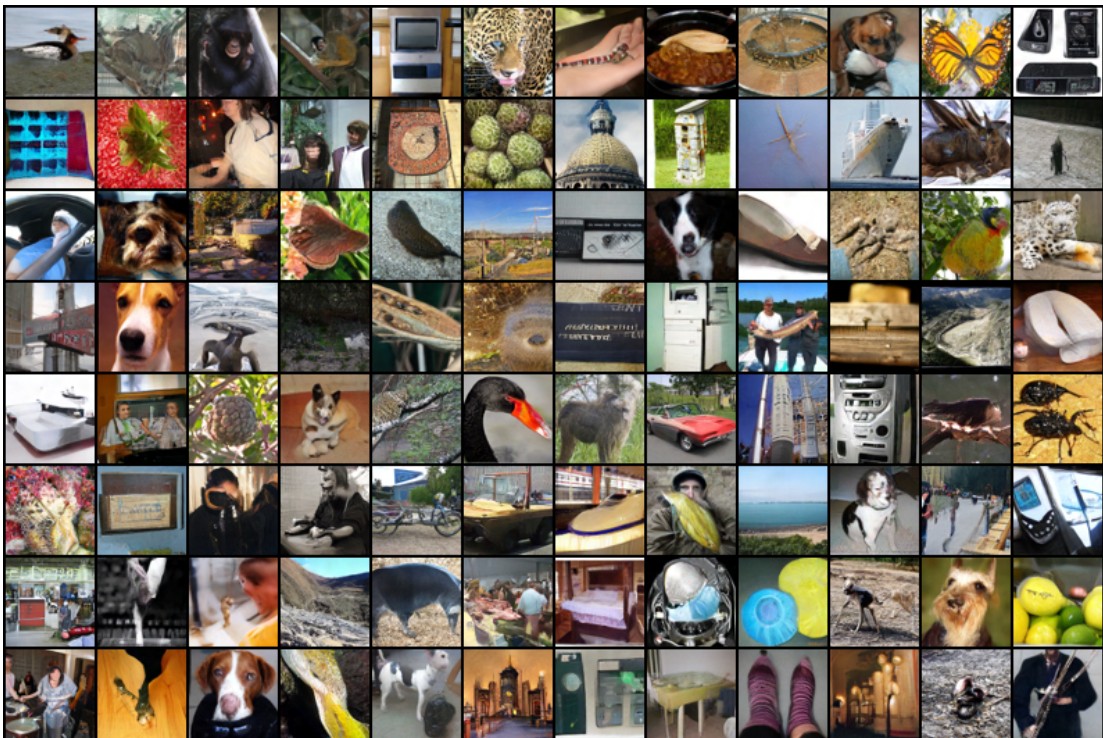

*(a)* Samples from CM trained via ECT.

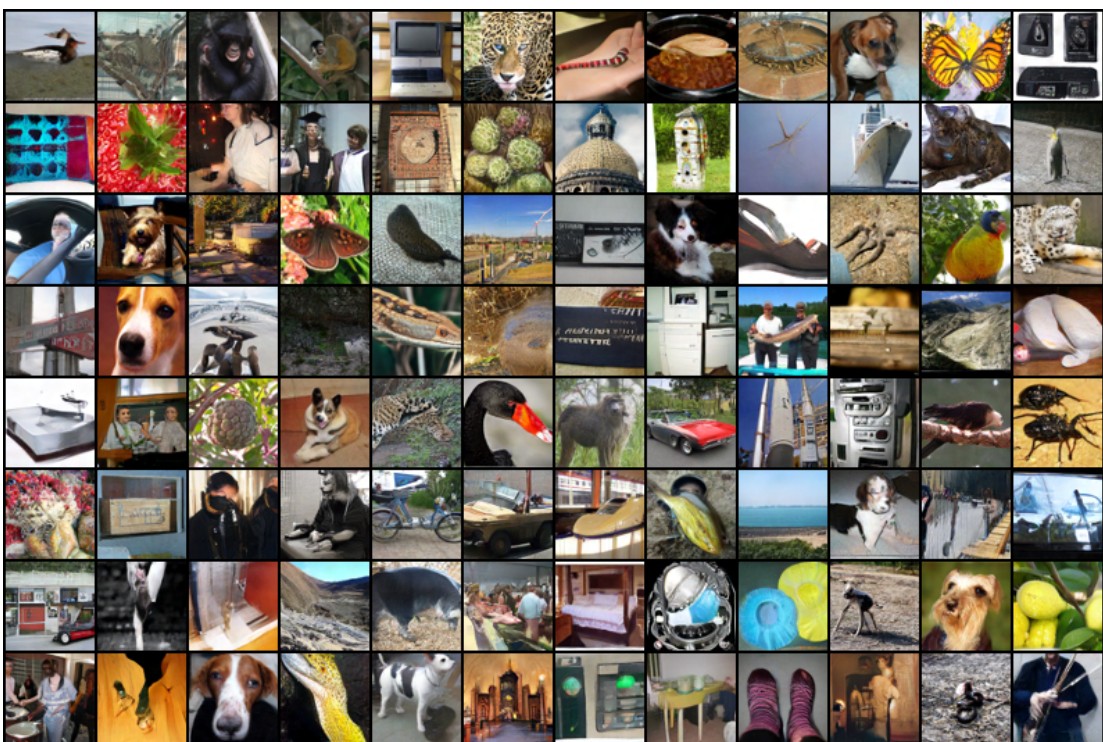

*(b)* Samples from CM trained via AYT (Ours).

*Figure 10.* Uncurated one-step CM samples on ImageNet $64 \times 64$.

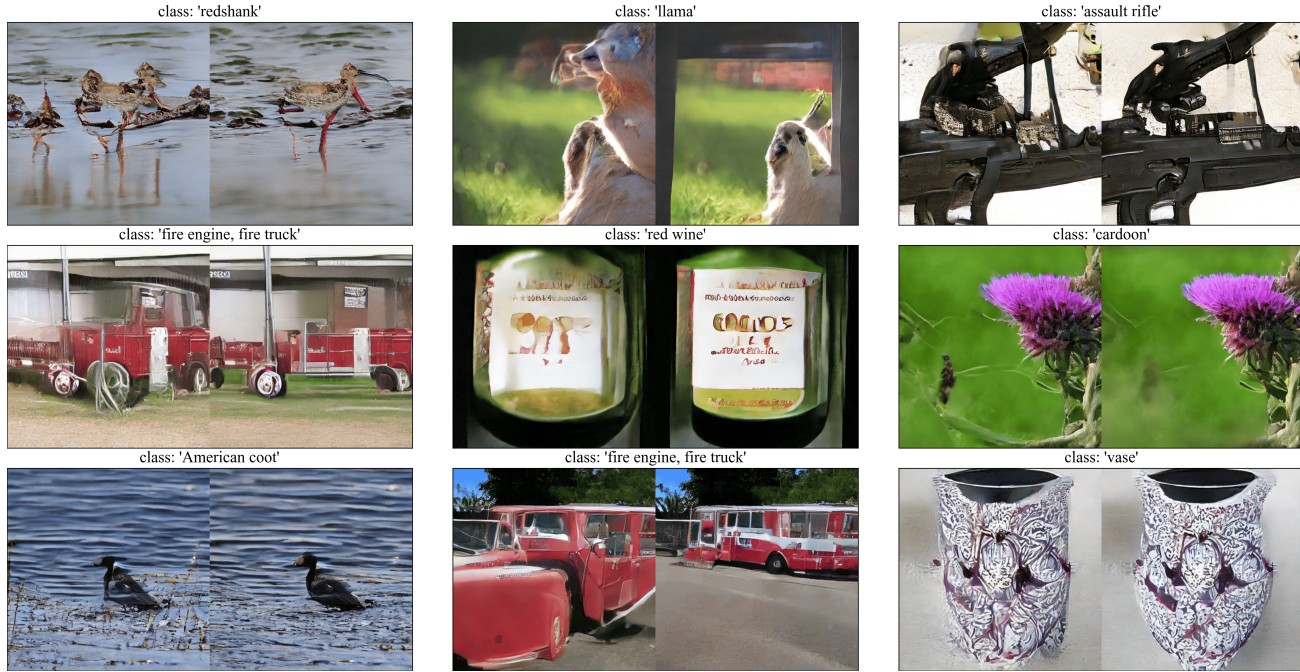

*Figure 11.* Curated one-step samples for ECT (left) and AYT (right) on ImageNet $512 \times 512$.

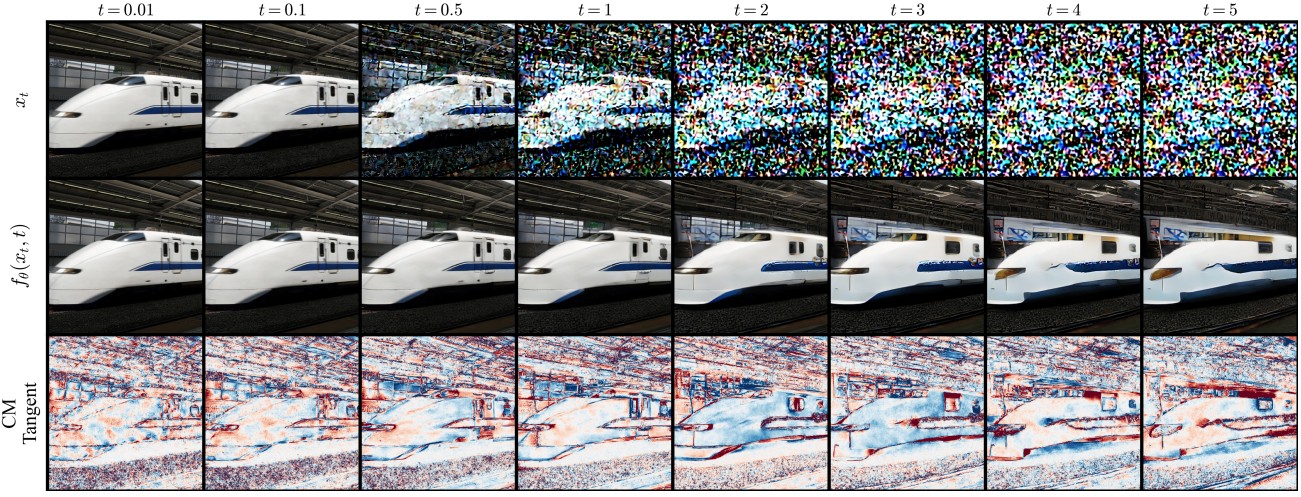

*Figure 12.* CM trajectory tangent visualization on ImageNet $512 \times 512$

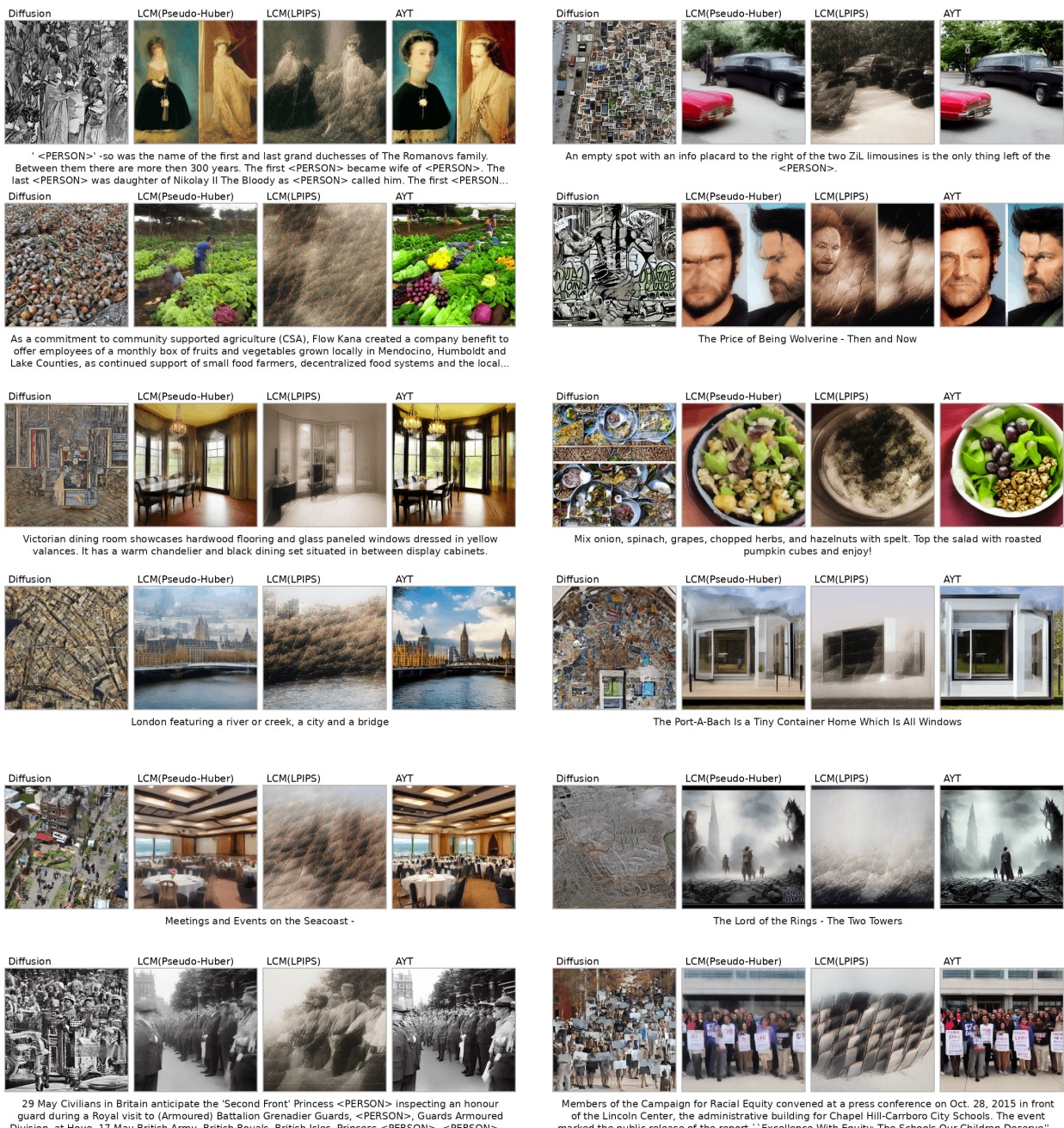

*Figure 13.* One-step text-to-image samples on CC12M comparing Diffusion and consistency distillation with different losses: Pseudo-Huber, LPIPS, and AYT. Each row uses the same text prompt shown below the generated samples.

