# OpenReview forum: "Align Your Trajectory Tangent: Training Better Consistency Models via Manifold-Aligned Tangents"
_ICML.cc/2026/Conference — ICML 2026 regular_

### Official Review · Reviewer_47nR · 2026-02-28

**Soundness:** 3
**Presentation:** 2
**Significance:** 2
**Originality:** 2
**Overall Recommendation:** 4
**Confidence:** 4

**Summary:**

This paper investigates the unstable training dynamics of consistency models (CMs) by analyzing their behavior through the lens of **trajectory tangents**, which describe how CM outputs evolve along probability‑flow ODE trajectories during training. The authors empirically observe that, near convergence, CM trajectory tangents exhibit strong oscillatory components that lie **parallel to the data manifold**, rather than pointing orthogonally toward it. They hypothesize that these manifold‑parallel components hinder efficient convergence. To address this, the paper proposes **Align Your Trajectory Tangent (AYT)**. The key idea is to compute consistency losses in a learned feature space whose gradients are designed to point toward the data manifold, thereby suppressing oscillatory, manifold‑parallel updates. The feature space is learned in a self‑supervised manner using synthetic image transformations without human supervision or class labels. Experiments on CIFAR‑10 and ImageNet‑64 show that AYT accelerates CM convergence, improves FID compared to standard pixel losses and LPIPS.

**Compliance With Llm Reviewing Policy:**

Affirmed.

**Final Justification:**

The rebuttal satisfactorily addressed my concerns regarding high-resolution evaluations, the causal relationship between manifold-parallel motion and slow convergence, and the tangent decomposition on real-world image manifolds. The paper is insightful, and the authors convincingly demonstrate the generalization and real-world applicability of their method.

**Key Questions For Authors:**

- Since all experiments are limited to CIFAR‑10 and ImageNet‑64, can the authors demonstrate that LPIPS genuinely harms CM training at higher resolutions where perceptual metrics are known to be more reliable?

- Is there any quantitative or qualitative evidence that the learned features meaningfully approximate orthogonal distance to the data manifold?

- When accounting for encoder training, memory usage, and per‑iteration overhead, does AYT still offer meaningful wall‑clock speedups compared to standard CM training?

- The synthetic experiments are highly favorable and low‑dimensional; is the same geometric interpretation meaningful for complex, high‑dimensional data?

**Limitations:**

Yes, the Limitations section says that their study has so far focused on relatively small-scale settings.

**Strengths And Weaknesses:**

### Strengths

- **Interesting diagnostic perspective.** The paper introduces trajectory tangents as a lens to analyze CM training dynamics, and the decomposition into manifold‑parallel and orthogonal components is conceptually interesting.
- **Clear empirical observation.** Visualizations convincingly show that CM trajectory tangents remain highly structured and oscillatory even near convergence, which is a non‑trivial and useful observation.
- **Self‑supervised alternative to LPIPS.** The proposed manifold feature distance avoids human supervision and ImageNet labels, which is appealing in principle.
- **Consistent empirical gains in the tested regime.** On CIFAR‑10 and ImageNet‑64, AYT improves convergence speed and FID relative to pixel loss and LPIPS under the authors’ setup.

### Weaknesses

- **Unfair or under‑supported criticism of LPIPS.** The claim that LPIPS degrades FID due to representation mismatch is only evaluated on low‑resolution datasets (CIFAR‑10, ImageNet‑64), where LPIPS is known to be less reliable. No higher‑resolution experiments or resolution‑controlled ablations are provided, making this criticism speculative.
- **Additional encoder introduces cost and uncertainty.** AYT requires training a separate VGG‑based manifold feature encoder. The paper does not quantify training cost, or sensitivity to feature quality, nor does it evaluate how accurately the learned scalar functions approximate true manifold distance.
- **Conceptual overreach regarding manifold‑parallel components.** The paper implicitly treats manifold‑parallel trajectory components as undesirable oscillations, but the causal link between manifold‑parallel motion and slow convergence is suggested visually but not rigorously established.
- **Limited generalization evidence.** Synthetic manifold experiments are highly favorable and low‑dimensional, and it is unclear whether the same tangent decomposition is meaningful for complex real‑world image manifolds.
- **Efficiency claims are iteration‑based.** Claims of “orders of magnitude” acceleration ignore encoder training and per‑iteration overhead, making the practical efficiency gains unclear.

---

> ### Author Rebuttal · Authors · 2026-03-30
>
> We thank Reviewer 47nR for the perceptive comments. We address your concerns below.
>
> **[W1] Higher-resolution experiments with LPIPS vs. AYT.**
>
> We further evaluate text-to-image Consistency Distillation (CD) using a pre-trained Stable Diffusion 1.5 model. We follow the Latent CM setup with LoRA layers and train on the Conceptual Captions 12M dataset at 512×512 resolution.
>
> For LPIPS, we minimize the distance between adjacent-timestep samples after decoding from the latent space. However, decoder-induced artifacts significantly degrade performance compared to both the baseline and AYT, suggesting that LPIPS may be unsuitable in latent-space pipelines.
>
> Overall, AYT remains effective in high-resolution and practical settings.
>
> ||FID $\downarrow$|||Clip-Score|$\uparrow$||
> |-|-|-|-|:-:|-|-|
> |*steps =*|1|2|4|1|2|4|
> |Diffusion|429.52|402.12|39.79|19.54|22.13|25.61|
> |Pseudo-Huber|61.11|20.87|13.33|25.93|29.32|30.23|
> |LPIPS|217.84|127.96|38.59|19.54|22.12|25.65|
> |AYT|50.61|15.39|11.53|26.31|29.65|30.34|
>
> *Tab. Stable Diffusion 1.5 CD.*
>
> **Sample link : https://anonymous.4open.science/r/icml26-rebuttal-2-6E4B/README.md**
>
> **[W2] Wall-clock time comparisons, sensitivity to feature quality, how accurately the learned scalar functions approximate manifold distance.**
>
> We refer the Reviewer to Tab. 3 in Append. B.3 where we ablate wall-clock efficiency and feature architecture (sensitivity to feature quality). We show AYT accelerates wall-clock time to convergence, even including encoder training and per-iteration overhead.
>
> We also provide a sanity check by training ECT with autoencoder (AE) features. This improves ECT but still falls short of AYT. This shows that feature-space training is beneficial, but the quality of the representation is critical -- naively learned features do not provide the same advantage as more principled constructions like AYT.
>
> ||1-step|2-step|
> |-|-|-|
> |ECT|3.48|2.13|
> |ECT + AE|3.08|2.34|
> |AYT|2.71|2.17|
>
> *Tab. FIDs for ECT with AE feature, CIFAR-10.*
>
> We would also like to clarify a potential misunderstanding -- we remind the Reviewer that manifold features do not approximate orthogonal distance to the data manifold. They approximate level sets of perturbed manifolds, so their gradients are manifold-orthogonal (Sec. 5.2). We also refer the Reviewer to Append. D, where we prove that optimal manifold feature gradients are manifold-orthogonal, validating AYT.
>
> **[W3] Link between manifold-parallel motion and slow convergence.**
>
> Consider the stochastic gradient variance
>
> $Var(\nabla_\theta (f_\theta^\top sg[g_\phi]))=(\nabla_\theta f_\theta)^\top Cov(g_\phi) (\nabla_\theta f_\theta):=M_\phi$.
>
> By Thm. 4.8 of [1], $\mathbb{E}||\nabla_\theta L(\theta)||^2\leq O(M_\phi)$, implying that larger gradient variance slows convergence. The convergence rate is thus governed by the covariance of the trajectory tangent $g_\phi$, modulated by $\nabla_\theta f_\theta$.
>
> In CM, the trajectory tangent is $g_I$, where $I$ is the identity. The tables below show that AYT consistently exhibits lower gradient variance than CM, which explains its faster convergence.
>
> We hypothesize that this difference arises from the structure of the trajectory tangents: oscillatory CM trajectory tangents $g_I$ induce a diffuse covariance matrix, while manifold-orthogonal AYT trajectory tangents yield a more concentrated covariance matrix. This produces lower-variance gradients and accelerates optimization.
>
> ||enc32|enc16|enc8|dec8|dec16|dec32|
> |-|-|-|-|-|-|-|
> |$M_I$|2.3e-5|1.1e-5|7.3e-6|1.3e-7|4.0e-7|5.1e-8|
> |$M_\phi$|1.2e-6|6.4e-7|3.3e-7|5.2e-9|6.7e-9|1.5e-9|
>
> *Tab. Gradient variance, CIFAR-10.*
>
> ||down0|down1|down2|up0|up1|up2|
> |-|-|-|-|-|-|-|
> |$M_I$|9.95e-1|2.47e-1|2.10e-1|5.63e-3|8.30e-2|6.31e-2|
> |$M_\phi$|2.79e-1|1.64e-1|1.43e-1|4.29e-3|6.41e-2|4.63e-2|
>
> *Tab. Gradient variance, LCM with SD 1.5.*
>
> **[W4] Tangent decomposition for real-world image manifolds.**
>
> To examine whether oscillatory trajectory tangents arise in higher dimensional datasets, we repeated the experiment from Fig. 4 of our paper on CIFAR-10 using continuous CD. Since the true CIFAR-10 data manifold is unknown, we approximate it by training an autoencoder. The Jacobian of the decoder spans the tangent space of the learned manifold (see, e.g., the proof of Thm. 2 in [2]).
>
> Consistent with results on the synthetic 2D discs dataset, CM trajectory tangents on CIFAR-10 exhibit non-trivial manifold-parallel components that grow with $t$, while AYT trajectory tangents remain predominantly manifold-orthogonal.
>
> |$t =$|0.01|0.1|1|10|80|
> |-|-|-|-|-|-|
> |CM|0.97|0.88|0.72|0.69|0.71|
> |AYT|0.98|0.96|0.94|0.93|0.94|
>
> *Tab. Fraction of manifold-orthogonal components in trajectory tangents for continuous CD.*
>
> [1] Large Scale Diffusion Distillation via Score-Regularized Continuous-Time Consistency
>
> [2] Manifold Preserving Guided Diffusion

---

> > ### Author Rebuttal · Reviewer_47nR · 2026-04-02
> >
> > Thank the authors for their detailed and thoughtful response, as well as the thorough additional experiments. They have fully addressed my concerns, particularly regarding high-resolution evaluations, the causal link between manifold-parallel motion and slow convergence, and the tangent decomposition on real-world image manifolds. I am pleased to raise my ratings and kindly encourage the authors to incorporate these results into the final version of the paper.

---

> > > ### Author Response · Authors · 2026-04-02
> > >
> > > We truly appreciate your thorough and thoughtful feedback, and we are glad to hear that your concerns have been fully resolved. Thank you as well for your encouraging remarks on our additional experiments and analyses.
> > > We will incorporate these results into the final version to further strengthen the paper.

---

### Official Review · Reviewer_Ezin · 2026-03-13

**Soundness:** 3
**Presentation:** 3
**Significance:** 3
**Originality:** 3
**Overall Recommendation:** 4
**Confidence:** 5

**Summary:**

This paper examines the training dynamics of consistency models near convergence and discovers that CM trajectory tangents are oscillatory--it moves parallel to the data manifold, rather than the expected towards the manifold. The authors then propose manifold feature distance (MFD) as the new loss function, which mitigates the oscillation and makes the tangents point toward the data manifold. MFD can accelerate the training of CMs, and even out-perform the LPIPS loss.

**Compliance With Llm Reviewing Policy:**

Affirmed.

**Final Justification:**

The problem of trajectory oscillation is interesting and never been discovered before. And the authors provide the loss function with good intuition and empirical studies. The new loss function also provides obvious improvement. For my primary concerns, the authors addressed them in the rebuttal--though I think theoretical analysis would make the paper stronger, the current empirical study and analysis also support the idea. As a result, I keep my positive rating 4.

**Key Questions For Authors:**

- Can you explain why CM oscillation happens, e.g., in mathematics and observations, experiments?

- Can you explain why the 2-step performance is worse than baseline ECT on both CIFAR-10 and ImageNet-64?

**Limitations:**

yes

**Strengths And Weaknesses:**

Strengths:

- The phenomenon discussed in the paper, i.e., when the training is near convergence, the CM tangents are parallel to rather than toward the manifold, is interesting and never been discovered before.

- The proposed method is well-motivated.

- The proposed new loss largely accelerates training, and achieves even better performance.

- The figures are well-made to support the ideas and provide better understanding for readers.

Weaknesses:

- The paper provide the observation of CM oscillation, but does not provide analysis of why this happens.

- The two-step performance in Table 1 underperforms the baseline ECT.

---

> ### Author Rebuttal · Authors · 2026-03-30
>
> We thank Reviewer Ezin for the essential feedback. We address your concerns below.
>
> **[W1] Why trajectory oscillation happens.**
>
> [1,2] attribute the slow training of CMs to error accumulation when propagating targets from $t=0$. To isolate sources of error, we consider continuous Consistency Distillation (CD), which removes conditional–marginal velocity mismatch and ODE discretization error. The remaining errors are (i) stochastic gradient noise and (ii) model approximation error.
>
> The table below shows that oscillatory trajectory tangents persist in continuous CD, with the manifold-parallel component increasing with $t$, while AYT trajectory tangents remain largely manifold-orthogonal. Since this experiment is conducted with a fixed model, this suggests that model approximation error is a primary source of manifold-parallel trajectory tangents.
>
> |$t =$|0.01|0.1|1|10|80|
> |-|-|-|-|-|-|
> |CM|0.97|0.88|0.72|0.69|0.71|
> |AYT|0.98|0.96|0.94|0.93|0.94|
>
> *Tab. Fraction of manifold-orthogonal components in continuous CD trajectory tangents, CIFAR-10.*
>
> The table below shows that smaller batch sizes degrade performance, indicating that gradient noise amplifies accumulated error. We thus hypothesize that stochastic gradient noise -- together with neural net inductive bias -- drives oscillatory trajectory tangents, and AYT mitigates them.
>
> ||1-step|2-step|
> |-|-|-|
> |CD, bs=64|5.32|4.33|
> |CD, bs=32|5.51|4.31|
> |CD + AYT, bs=64|3.90|3.65|
> |CD + AYT, bs=32|4.09|3.85|
>
> *Tab. FIDs on continuous CD, CIFAR-10.*
>
> **[W2] The two-step performance in Table 1 underperforms the baseline ECT.**
>
> We first remind the Reviewer that the drop in two-step performance is negligible relative to the substantial improvement observed in the one-step setting.
>
> Furthermore, as shown in Table 2 (Appendix B.2), AYT outperforms ECT across several additional metrics (e.g., Inception Score on CIFAR-10, Inception Score, Recall and Density on ImageNet with 2-step samples), suggesting improved sample diversity. We therefore believe that the comparison should be based on a broader set of metrics and evaluation settings, rather than a single metric in a single setting.
>
> We further hypothesize that the slight discrepancy in two-step performance is not due to a limitation of our method, but rather stems from external factors such as the limited dataset size, the complexity of the EDM2 architecture, and the stabilization techniques required for ImageNet-64 training.
>
> To support this claim, we further evaluate text-to-image Consistency Distillation (CD) using a pre-trained Stable Diffusion 1.5 model. We follow the Latent CM setup with LoRA layers and train on the Conceptual Captions 12M dataset at 512×512 resolution. As shown below, AYT consistently outperforms the baseline across all 1,2,4 step settings.
>
> ||FID $\downarrow$|||Clip-Score|$\uparrow$||
> |-|-|-|-|:-:|-|-|
> |*steps =*|1|2|4|1|2|4|
> |Diffusion|429.52|402.12|39.79|19.54|22.13|25.61|
> |Pseudo-Huber|61.11|20.87|13.33|25.93|29.32|30.23|
> |LPIPS|217.84|127.96|38.59|19.54|22.12|25.65|
> |AYT|50.61|15.39|11.53|26.31|29.65|30.34|
>
> *Tab. Stable Diffusion 1.5 Consistency Distillation.*
>
> **Sample link : https://anonymous.4open.science/r/icml26-rebuttal-2-6E4B/README.md**
>
> [1] Consistency Model Made Easy
>
> [2] Large Scale Diffusion Distillation via Score-Regularized Continuous-Time Consistency
>
> [3] Manifold Preserving Guided Diffusion

---

> > ### Author Rebuttal · Reviewer_Ezin · 2026-04-04
> >
> > I thank the authors for providing detailed response and experiments in the rebuttal, which addressed my concerns. I will keep my rating 4, which recommends acceptance. Though I think a theoretical analysis on trajectory oscillation would be better, the current empirical results are also sufficient to support the idea.

---

> > > ### Author Response · Authors · 2026-04-04
> > >
> > > We sincerely thank you for your detailed and insightful feedback, and we are pleased to know that your concerns have been addressed. We also greatly appreciate your positive comments regarding our additional experiments and analyses. These results will be included in the next version to further enhance and reinforce the quality of the paper.

---

### Official Review · Reviewer_8zQE · 2026-03-13

**Soundness:** 3
**Presentation:** 4
**Significance:** 3
**Originality:** 3
**Overall Recommendation:** 5
**Confidence:** 4

**Summary:**

This paper identifies that consistency models suffer from so-called 'trajectory tangents' (total derivative of the network w.r.t. diffusion time) having components lying parallel to the data manifold, whereas they should ideally point towards the data manifold. It is assumed that this non-alignment of trajectory tangents causes instability or oscillations in the training of consistency models.

To solve this, this paper proposes to compute the consistency loss in a latent space designed to provably eliminate these manifold-parallel directions with an encoder predicting the intensity of various image perturbations / augmentations (such as blurring and rotating). Conducted experiments show that this 'AYT' loss improves consistency model generation performance (w.r.t. standard Huber and LPIPS consistency losses), and stabilizes and accelerates training.

**Compliance With Llm Reviewing Policy:**

Affirmed.

**Final Justification:**

As explained above, this is a good paper proposing a novel perspective on consistency model training, with an insightful analysis followed by a successful and an interesting solutions. The flaws I initially pointed our were largely addressed in the rebuttal with new experiments and elements of discussion. I believe this paper deserves to be published at ICML.

**Key Questions For Authors:**

This is an interesting, well-written paper with good impact potential. Most of the main claims are supported even though that could be improved. Therefore, I recommend a weak accept for the time being and am willing to raise my grade depending on the authors' response. I expect in particular the following, according to the weaknesses listed above.
1. Clarify or better support claims on the origin and consequences of manifold-parallel components in the tangents.
2. Add/discuss the requested comparisons.
3. Develop part of the experiments listed above.
4. Improve, if relevant, the explanation of trajectory tangents.
5. Discuss the surprising effectiveness of a consistency loss in a low-dimensional latent space.

---

**Post-rebuttal update.** The author's response addressed all my concerns and pointed out some details I missed in my initial assessment. As a result and provided that the new elements are well integrated in the final version, I recommend to accept this paper. Accordingly, I raised the soundness score from 2 to 3, the presentation score from 3 to 4 and the overall recommendation from Weak accept to Accept.

**Limitations:**

Yes

**Strengths And Weaknesses:**

### Strengths

1. The approach and solution are **novel**. The paper provides a new perspective on improving consistency model training that, to my knowledge, has not been studied in the literature.
2. To my understanding, **the analysis is sound and the merits of the approach are proven**: they are supported by a convincing theory and promising experiments showing robust performance improvements.
3. The experimental analysis identifying on-manifold / off-manifold directions provides **relevant insights** into consistency models, and the AYT improvement an **elegant solution** to the raised problem.
4. The focus on the trajectory tangents and their alignment toward the data manifold is of interest to most researchers in the area of generative models with flow maps as it may be even be applicable beyond consistency models to other generative models with similar training losses (Franceschi et al., 2023; Deng et al., 2026). As such, I think this paper has the **potential to influence further work** in this research area.
5. The paper is **well written** and easy to read -- with a few exceptions detailed in the weaknesses. I think it is accessible, which will maximize its potential impact.


### Weaknesses

1. A few secondary claims regarding the origin of non-aligned tangents in the analysis **lack support or would benefit from further analysis**.
   - The paper mentions multiple times that non-aligned tangents can cause oscillatory behaviors or slow convergence during training. While manifold-parallel components can suggest this, there is no empirical or theoretical support, especially as it is unclear how trajectory tangents translate into parameter updates.
   - The paper also suggests that these manifold-parallel components may arise from training stochasticity. I would advise to conduct experiments similar to Section 4 with varying batch sizes and in the consistency distillation setting in order to assess how the main sources of stochasticity affect the tangents.
   - Qualitative experiments in Section 4 on CIFAR-10 do not clearly explain why structured patterns suggest oscillatory trajectories; this is better understood in a second read by comparing them to the results on the synthetic dataset. It is also unclear how "that could imply large movements along the manifold" (l. 198).
2. The paper should include **additional comparisons** with related approaches to AYT, although I understand there are not many. I would suggest the following two.
   - Adversarial losses have already been used to improve consistency training (Kong et al., 2024). More generally, adversarial losses are known to steer the generated distribution towards the data distribution.
   - As a sanity check, a baseline with a standard autoencoder as $\phi$ could be considered.
3. A **few secondary experimental claims** could be improved.
   - The paper claims that AYT greatly accelerates training. It is true when focusing on the time at which AYT achieves the baseline's performance, but I would argue that the baseline may never reach the performance level of AYT. The acceleration claim would be stronger if, at equal performance when converged, AYT achieves it faster. As is, it looks like from plots in Figures 5 and 6 that AYT simply performs better that the baseline ECT (which is even better than acceleration).
   - The batch size ablation should also present varying batch sizes for the baseline ECT, to assess whether the performance loss is similar to or worse than AYT's.
   - Main experiments are only performed on smaller models (S versions of EDM/ECT). Experiments on larger (L/M versions) would provide stronger experimental support. However, I understand that this requires more compute which not every researcher can afford. Less importantly, stronger improvements on ImageNet 512 would also be beneficial, but ECT's baseline results on this dataset are difficult to reproduce so I would not read too much into it.
4. The **explanation of the role of trajectory tangents**, lines 198-208 in Section 3, is quite obscure. I better understand the tangents as the directions towards which outputs of $f_\theta$ are pushed during training, if the loss is reformulated (cf. Weber (2023), Eq. (26)-(31)). The benefits of having tangents directed towards the data manifold would be clearer explained this way.
5. Not a strong weakness given the experimental results but still a concern, there is no explanation why **consistency loss in such a small latent space** ($n=15$) can yield such strong results. This is surprising: even though gradients are projected in the right direction, the capacity loss is important. This should be discussed somewhere in the paper.

Deng et al. Generative Modeling via Drifting. arXiv:2602.04770, 2026.\
Franceschi et al. Unifying GANs and Score-Based Diffusion as Generative Particle Models. NeurIPS 2023.\
Kong et al. ACT-Diffusion: Efficient Adversarial Consistency Training for One-step Diffusion Models. CVPR 2024.\
Weber. The Score-Difference Flow for Implicit Generative Modeling. TMLR, 2023.

----

The following are minor flaws that do not impact my overall recommendation.
- The hypothesis in Section 6.1 that LPIPS divergence on CIFAR-10 is due to distribution mismatch with ImageNet could be confirmed by conduction the same experiment on ImageNet.
- The use of "overwhelmingly" l. 273 is excessive.
- The authors state that models are trained "from scratch" in Section 6.2. Does this mean that they do not use a pretrained diffusion model at initialization?
- Formatting:
  - I suspect $s$ should be $t'$ in Eq. (6).
  - In-sentence citations should be without parentheses (e.g. in Section 5.2).

---

> ### Author Rebuttal · Authors · 2026-03-30
>
> We thank Reviewer 8zQE for the thoughtful comments. We address your concerns below. Minor issues will be fixed accordingly.
>
> **[W1] Sources of manifold-parallel trajectory tangents.**
>
> [1,2] attribute slow CM training to error accumulation when propagating targets from $t=0$. To isolate sources of error, we consider continuous Consistency Distillation (CD), removing conditional–marginal velocity mismatch and ODE discretization error. The remaining errors are (i) stochastic gradient noise and (ii) model approximation error.
>
> Below we observe oscillatory trajectory tangents persist in continuous CD with larger $t$. Here the model is fixed, so this suggests model approximation error is a primary source of manifold-parallel trajectory tangents.
>
> |$t =$|0.01|0.1|1|10|80|
> |-|-|-|-|-|-|
> |CM|0.97|0.88|0.72|0.69|0.71|
> |AYT|0.98|0.96|0.94|0.93|0.94|
>
> *Tab. Fraction of manifold-orthogonal components in continuous CD trajectory tangents, CIFAR10.*
>
> Below, smaller batch sizes degrade performance, indicating gradient noise amplifies accumulated error. We thus hypothesize that stochastic gradient noise, with neural net inductive bias, drives oscillatory trajectory tangents, and AYT mitigates them.
>
> ||1-step|2-step|
> |-|-|-|
> |CD, bs=64|5.32|4.33|
> |bs = 32|5.51|4.31|
> |CD + AYT, bs=64|3.90|3.65|
> |bs = 32|4.09|3.85|
>
> *Tab. FIDs on continuous CD, CIFAR10.*
>
> **[W2] How non-aligned trajectory tangents can cause slow convergence.**
>
> Consider the stochastic gradient variance
>
> $Var(\nabla_\theta (f_\theta^\top sg[g_\phi]))=(\nabla_\theta f_\theta)^\top Cov(g_\phi) (\nabla_\theta f_\theta):=M_\phi$.
>
> By Thm. 4.8 of [3], $\mathbb{E}||\nabla_\theta L(\theta)||^2\leq O(M_\phi)$, implying that larger gradient variance slows convergence. The convergence rate is thus governed by the covariance of the trajectory tangent $g_\phi$, modulated by $\nabla_\theta f_\theta$.
>
> In CM, the trajectory tangent is $g_I$, where $I$ is the identity. Tables below show AYT consistently exhibits lower gradient variance than CM, which explains its faster convergence.
>
> We hypothesize that this difference arises from the structure of the trajectory tangents: oscillatory CM $g_I$ induce a diffuse covariance matrix, while manifold-orthogonal AYT $g_\phi$ yield a concentrated covariance. This produces lower-variance gradients and accelerates optimization.
>
> ||enc32|enc16|enc8|dec8|dec16|dec32|
> |-|-|-|-|-|-|-|
> |$M_I$|2.3e-5|1.1e-5|7.3e-6|1.3e-7|4.0e-7|5.1e-8|
> |$M_\phi$|1.2e-6|6.4e-7|3.3e-7|5.2e-9|6.7e-9|1.5e-9|
>
> *Tab. Gradient variance, CIFAR10.*
>
> ||down0|down1|down2|up0|up1|up2|
> |-|-|-|-|-|-|-|
> |$M_I$|9.95e-1|2.47e-1|2.10e-1|5.63e-3|8.30e-2|6.31e-2|
> |$M_\phi$|2.79e-1|1.64e-1|1.43e-1|4.29e-3|6.41e-2|4.63e-2|
>
> *Tab. Gradient variance, LCM with SD 1.5.*
>
> **[W3] Paper Fig. 2 -- why structured patterns suggest oscillatory trajectories.**
>
> Tangent space vectors at an image are local factors of variation on the manifold, and thus should point in directions which move toward other images. This led us to hypothesize that structured patterns indicate presence of manifold-parallel components in trajectory tangent. We verify that this is indeed true in answer to [W1].
>
> **[W4] Additional experiments.**
>
> *Large-scale T2I.* With Stable Diffusion (SD) 1.5 T2I CD (512×512), AYT consistently beats others, showing its scalability.
>
> ||FID $\downarrow$|||Clip-Score|$\uparrow$||
> |-|-|-|-|:-:|-|-|
> |*steps =*|1|2|4|1|2|4|
> |Diffusion|429.52|402.12|39.79|19.54|22.13|25.61|
> |Pseudo-Huber|61.11|20.87|13.33|25.93|29.32|30.23|
> |LPIPS|217.84|127.96|38.59|19.54|22.12|25.65|
> |AYT|50.61|15.39|11.53|26.31|29.65|30.34|
>
> *Tab. T2I results.*
>
> **Sample link : https://anonymous.4open.science/r/icml26-rebuttal-2-6E4B/README.md**
>
> *ECT ablations.*
>
> - Batch size: refer to paper Fig. 1
> - Longer training: 800k training steps yields marginal gains; matching AYT would require impractical scaling.
> - AE features: improves over ECT but still below AYT, showing feature quality is critical.
> - Adversarial: faster early training but overfits and worsens final FID; stronger discriminators are typically required.
>
> ||1-step|2-step|
> |-|-|-|
> |ECT (400k iters)|3.48|2.13|
> |ECT (800k iters)|3.15|2.15|
> |ECT + AE (400k iters)|3.08|2.34|
> |ECT + Adv. (400k iters)|3.63|2.19|
> |AYT (400k iters)|2.71|2.17|
>
> *Tab. ECT ablation FIDs, CIFAR10.*
>
> **[W5] Explanation of the role of trajectory tangents is obscure.**
>
> Eq. (7) is equivalent to
>
> $\min_\theta \mathbb{E}[||f_\theta(x_t,t)-sg[(f_\theta(x_t,t)-\eta d f_\theta(x_t,t) / dt)]||^2]$
>
> showing CM pushes outputs along trajectory tangents. If this is better, we will replace Eq. (7) with above.
>
> **[W6] Why consistency loss in such a small latent can yield strong results.**
>
> We also use intermediate features (lines 288-290 right), explaining the strong performance. Fig. 4 shows these also yield manifold-orthogonal trajectory tangents.
>
> [1] ECT
>
> [2] Large Scale Diffusion Distillation via Score-Regularized Continuous-Time Consistency
>
> [3] Optimization Methods for Large-Scale Machine Learning

---

> > ### Author Rebuttal · Reviewer_8zQE · 2026-04-02
> >
> > I would like to thank the authors for their thorough response, which adequately addressed all my concerns. The strong new experiments and the additional explanations improve the support of the claims and the presentation quality. Provided that they are integrated in the final version of the paper, I am happy to fully recommend the paper acceptance.
> >
> > As for "[W5] Explanation of the role of trajectory tangents is obscure.", I do think this equation illustrates well the role of the tangents. I would advise the authors to keep their original Eq. (7) that is more widely known and complement it with the newly proposed equation in their response.

---

> > > ### Author Response · Authors · 2026-04-02
> > >
> > > We’re pleased to hear that your earlier concerns have been resolved. We sincerely appreciate your thoughtful feedback and the time you devoted to reviewing our work, and we will carefully incorporate all points from this discussion into the next revision of our manuscript.

---

### Official Review · Reviewer_FuBh · 2026-03-13

**Soundness:** 3
**Presentation:** 3
**Significance:** 2
**Originality:** 3
**Overall Recommendation:** 4
**Confidence:** 4

**Summary:**

The paper proposes manifold feature distance (MFD) to align the tangent of consistency models with the data manifold and accelerate training. Specifically, a feature map is learned by predicting the level of certain types of corruptions, whose gradient produces manifold-othogonal directions and filters the consistency tangent when used in the consistency loss. Experiments show that the method can accelerate and stablize CM training on CIFAR10 and ImageNet 64×64 datasets.

**Compliance With Llm Reviewing Policy:**

Affirmed.

**Final Justification:**

The rebuttal resolved most of my concerns. I hope the authors could apply the method to larger-scale settings in the future.

**Key Questions For Authors:**

- Could the authors provide more results on consistency distillation, continuous-time CMs and text-to-image or text-to-video tasks? I would strongly support this paper if the authors could sufficiently show the method's scalability.
- Or alternatively, are there any training-free choices of the feature map that could be plugged into continuous-time CMs and improve the training speed and stability? This way it can be directly applied to hard text-to-video tasks.

**Strengths And Weaknesses:**

Strengths:
- The theoretical analyses provide interesting intuition on how a feature map used in consistency loss could help align the tangent with manifold orthogonal.
- Experiments demonstrate the effectiveness in accelerating and stablizing CM training, and advantages compared to off-the-shelf feature extractor LPIPS.

Weaknesses:
- It is unclear whether the manifold alignment issues still exist in consistency distillation, where a teacher provides much cleaner signals, and whether the proposed method is still effective. Consistency distillation is more important than consistency training in large-scale tasks.
- Though the author conduct theoretical analyses from the continuous-time tangent perspective, there are no results on continuous-time CMs [1].
- Experiments are limited to small-resolution toy dataset, lacking results on larger and common datasets like ImageNet 256x256. The shift from ImageNet 64x64 to ImageNet 256x256 could result in quite different outcomes. As far as I know, the main baseline method used in this paper, ECT, does not perform well when scaled up to ImageNet 256x256, not as good as iCT.
- The scability issue is a larger concern if considering diffusion in real application scenarios. In rCM [2], it is empirically observed that CMs on strong-conditioned text-to-image and text-to-video tasks could exhibit significant quality flaws, and FID on academic datasets is no longer a good indicator. Considering that the method requires additional training of feature map network and design of perturbations, it is doubtful how it will perform on these tasks.
- Some minor typos, for example writting CM as consistency matching in Proposition 5.1.

[1] Simplifying, Stabilizing and Scaling Continuous-Time Consistency Models

[2] Large Scale Diffusion Distillation via Score-Regularized Continuous-Time Consistency

---

> ### Author Rebuttal · Authors · 2026-03-30
>
> We thank Reviewer FuBh for the insightful suggestions, and agree that scalability is important. We address your concerns and questions below. Minor typos will be fixed in the revised version of the paper.
>
> **[W1] Whether oscillatory trajectory tangents also arise in Consistency Distillation (CD) and Continuous CMs.**
>
> To examine whether oscillatory trajectory tangents arise in CD and continuous-time CMs, we repeated the experiment from Fig. 4 of our paper (trajectory tangent decomposition into manifold-parallel and orthogonal components) on CIFAR-10 using continuous CD (see the table below, where we report the fraction of manifold-orthogonal components in trajectory tangents for continuous CD, CIFAR-10).
>
> Since the true CIFAR-10 data manifold is unknown, we approximate it by training an autoencoder. The Jacobian of the decoder (input with respect to the latent) spans the tangent space of the learned manifold (see, e.g., the proof of Thm. 2 in [1]).
>
> Consistent with results on the synthetic two-dimensional discs dataset, CM trajectory tangents on CIFAR-10 exhibit non-trivial manifold-parallel components as $t$ grows, indicating the presence of oscillatory trajectory tangents. In contrast, AYT trajectory tangents remain predominantly manifold-orthogonal.
>
> |  | t=0.01 | 0.1 | 1 | 10 | 80 |
> |---|---|---|---|---|---|
> | CM | 0.97 | 0.88 | 0.72 | 0.69 | 0.71 |
> | AYT | 0.98 | 0.96 | 0.94 | 0.93 | 0.94 |
>
> *Tab. Fraction of manifold-orthogonal components in trajectory tangents for continuous CD.*
>
> **[W2] Results on Consistency Distillation (CD), continuous-time CMs and text-to-image or text-to-video tasks.**
>
> In the table below, we evaluate continuous CD on CIFAR-10 by taking the limit $\Delta t \to 0$ in ECD. AYT consistently outperforms CD by a significant margin, demonstrating that trajectory tangent alignment remains effective in continuous and distillation settings.
>
> |  | 1-step | 2-step |
> |---|---|---|
> | CD, bs=64 | 5.32 | 4.33 |
> | CD, bs=32 | 5.51 | 4.31 |
> | CD + AYT, bs=64 | 3.90 | 3.65 |
> | CD + AYT, bs=32 | 4.09 | 3.85 |
>
> *Tab. FIDs on continuous CD, CIFAR-10.*
>
> In the table below, we further evaluate text-to-image CD using a pre-trained Stable Diffusion 1.5 model. We follow the Latent CM setup with LoRA layers and train on the Conceptual Captions 12M dataset at 512×512 resolution.
>
> For LPIPS, we minimize the distance between adjacent-timestep samples after decoding from the latent space. However, decoder-induced artifacts significantly degrade performance compared to both the baseline and AYT. This suggests that LPIPS may be unsuitable in latent-space training pipelines.
>
> Overall, AYT remains effective in high-resolution and practical settings such as text-to-image generation.
>
> ||FID $\downarrow$|||Clip-Score|$\uparrow$||
> |-|-|-|-|:-:|-|-|
> |*steps =*|1|2|4|1|2|4|
> |Diffusion|429.52|402.12|39.79|19.54|22.13|25.61|
> |Pseudo-Huber|61.11|20.87|13.33|25.93|29.32|30.23|
> |LPIPS|217.84|127.96|38.59|19.54|22.12|25.65|
> |AYT|50.61|15.39|11.53|26.31|29.65|30.34|
>
> *Tab. Stable Diffusion 1.5 Consistency Distillation.*
>
> **Sample link : https://anonymous.4open.science/r/icml26-rebuttal-2-6E4B/README.md**
>
> [1] Manifold Preserving Guided Diffusion

---

> > ### Author Rebuttal · Reviewer_FuBh · 2026-04-04
> >
> > Thank you for the response. I am curious what "taking the limit $\Delta t \to 0$ in ECD" means. There is a non-trivial gap between truly continuous-time CMs (sCM), which rely on JVP, and discrete-time CMs with a very small $\Delta t$.
> >
> > Anyway, I appreciate the method's novelty and potential. I would like to raise my score. That being said, CIFAR-10 and SD1.5 text-to-image are very old settings and can not clearly reflect a method's effectiveness/scalability. I hope the authors could apply the method to sCM on ImageNet 512x512 [1] and Wan text-to-video settings [2] in the final paper, or in a follow-up paper.
> >
> > [1] https://github.com/NVlabs/FastGen
> > [2] https://github.com/NVlabs/rcm

---

> > > ### Author Response · Authors · 2026-04-04
> > >
> > > We sincerely appreciate your careful and constructive feedback, and we are pleased that your concerns have been addressed. We will incorporate this discussion into the final manuscript to further improve its clarity and overall quality.
> > >
> > > By “taking the limit $\Delta t \rightarrow 0$,” we refer to extending Easy Consistency Distillation (ECD) to the continuous setting by letting $\Delta t$ approach zero. In practice, this corresponds to using JVPs to compute the relevant terms in the loss, while keeping other hyperparameters—such as the time-dependent weighting and the sampling distribution over $t$—unchanged.
> > >
> > > Following the reviewer’s suggestion, we will also conduct additional experiments on ImageNet 512×512 and WAN T2V, and will report these results either in the final version of the paper or in a subsequent follow-up once they become available.

---

### Decision · Program_Chairs · 2026-04-30

**Decision:**

Accept (regular)

**Comment:**

This paper proposes a new way to train consistency models inspired by the observation that the trajectories of consistency models oscillate in directions parallel to the data manifold, rather than always pointing towards it. All the reviewers agreed the studied phenomenon is interesting, praised the novelty of the method and analysis, as well as the strength of the empirical results. Overall this paper is a clear accept.